

# Misidentified subglacial lake beneath the Devon Ice Cap, Canadian Arctic: A new interpretation from seismic and electromagnetic data

Siobhan F. Killingbeck[1,2], Anja Rutishauser[3], Martyn J. Unsworth[4], Ashley Dubnick[5], Alison S. Criscitiello[1], James Killingbeck[6], Christine F. Dow[2], Tim Hill[7], Adam D. Booth[8], Brittany Main[9], Eric Brossier[6]

[1]Department of Earth and Atmospheric Sciences, Faculty of Science, University of Alberta, Canada.
[2]Department of Geography & Environmental Management, Faculty of Environment, University of Waterloo, Canada.
[3]Geological Survey of Denmark and Greenland.
[4]Department of Physics, Faculty of Science, University of Alberta, Canada.
[5]YukonU Research Centre, Yukon University, 520 College Drive, Whitehorse, YT, Y1A 5N5.
[6]Independent.
[7]Department of Earth Sciences, Simon Fraser University, Canada.
[8]School of Earth and Environment, University of Leeds, UK.
[9]Department of Geography, Environment and Geomatics, University of Ottawa, Ottawa ON, Canada.

*Correspondence to*: Siobhan F. Killingbeck (skillin1@ualberta.ca)

**Abstract.** In 2018 the first subglacial lake in the Canadian Artic was proposed to exist beneath Devon Ice cap, based on the analysis of airborne radar data. Here, we report a new interpretation of the subglacial material beneath Devon Ice Cap, supported by data acquired from multiple surface-based geophysical methods in 2022. The geophysical data recorded included 9 km of active source seismic reflection profiles, 7 transient electromagnetic soundings and 17 magnetotellurics stations. These surface-based geophysical datasets were collected above the inferred locations of the subglacial lakes and show no evidence for the presence of subglacial water. The acoustic impedance of the subglacial material, estimated from the seismic data, is $9.49 \pm 1.92 \times 10^6$ kg m$^{-2}$ s$^{-1}$, comparable to consolidated or frozen sediment. The resistivity models obtained by inversion of both the transient electromagnetic and magnetotelluric measurements show the presence of highly resistive rock layers (1000 – 100000 Ω.m) directly beneath the ice. Re-evaluation of the airborne reflectivity data show that the radar attenuation rates were likely overestimated, leading to an overestimation of the basal reflectivity in the original radar studies. Here, we derive new radar attenuation rates using the temperature- and chemistry-dependent Arrhenius equation, and when applied to correct the returned bed power, the bed power does not meet the basal reflectivity threshold expected over subglacial water. Thus, the radar interpretation is now consistent with the seismic and electromagnetic observations of dry or frozen, non-conductive basal material.

## 1 Introduction

Analysis of radio echo sounding (RES) data acquired in 2011-2015 (Rutishauser et al., 2018) and 2018 (Rutishauser et al., 2022) proposed that the first subglacial lake in the Canadian Arctic had been detected beneath Devon Ice Cap (DIC) (Fig. 1a).





The proposed lake covered an area of 11.6 km$^2$ and was identified from a combination of higher relative basal RES reflectivity (proxy for dielectric contrast between two materials) and specularity content (proxy for wavelength scale roughness) over a

hydraulically flat region (Rutishauser et al., 2018; Rutishauser et al., 2022). These are characteristics that are consistent with the typical signature of subglacial lakes (e.g., Carter et al., 2007). The proposed lake was located in a trough beneath 760 m of ice near the summit of DIC (75°19'2.26"N 82°46'32.62"W), in a region where the ice was thought to be frozen to the bed (Van Wychen et al., 2017). Recent temperature modelling suggested cold basal temperatures estimated between -10°C and -14°C (Rutishauser et al., 2018). Therefore, without surface meltwater input the inferred subglacial lake beneath DIC required a high

salinity content to depress the freezing point and enable water to exist in its liquid form at the estimated cold basal temperatures. Thus, the lake was proposed to be hypersaline (Rutishauser et al., 2018), where geologic modelling suggested that the solute for the brine came from an underlying evaporite-rich sediment unit containing interbedded salt sequences, called the Bay Fiord Formation (Rutishauser et al., 2018).

        RES data is highly effective at mapping the ice base boundary and has excellent spatial coverage. However, as with

all geophysical methods, multiple interpretations can fit a RES data set. For example, anomalously strong and continuous basal reflections in RES data could indicate the presence of 1) subglacial water (Carter et al., 2010); (2) water-saturated and highly conductive sediment (Tulaczyk and Foley, 2020); or (3) smooth bedrock/sediments (Hofstede et al., 2023; Jordan et al., 2017). Furthermore, water usually contains significant dissolved salts and is electrically conductive. This results in significant attenuation of radar signals through the skin-depth effect. This means that reflections from structures at the bottom of lakes

are rarely observed in RES data, making it difficult to distinguish a subglacial lake from a layer of saturated sediments or a thin sheet of water at the glacier bed. Finally, difficulties in constraining radar attenuation rates can lead to misinterpretation of basal conditions (Matsuoka, 2011). The non-uniqueness of RES interpretations can be overcome with complementary geophysical surveys, such as active source seismic and electromagnetic (EM) methods. Seismic, EM and radar data are sensitive to different material properties of the subsurface, and the combination of the three methods offers more robust

evidence of subglacial water.

        Seismic methods provide acoustic properties of the ice base interface which can give independent material properties of the subsurface to confirm the presence of a subglacial lake. The acoustic impedance (product of density and compressional wave velocity) contrast across the ice base interface provides information on whether the material directly under the ice is acoustically soft (e.g., a lake, negative polarity reflection) or acoustically hard (e.g., consolidated sediment, positive polarity

reflection) relative to ice. Seismic methods are also capable of measuring lake depth since water is not attenuative to seismic energy. Therefore, where available, seismic evidence is key for diagnosing the presence and thickness of a subglacial lake.

        EM techniques, such as transient electromagnetics (TEM) and magnetotellurics (MT), measure the subglacial electrical resistivity structure, and are particularly applicable for glacial hydrological studies due to the large difference in electrical resistivity between ice ($10^4 - 10^8$ Ω.m) and water ($10^{-1} - 10^2$ Ω.m) (Key and Siegfried, 2017). Furthermore, the

electrical resistivity of water decreases rapidly with salinity (Killingbeck et al., 2021). However, EM methods are sensitive to the conductance, the product of conductivity (inverse of electrical resistivity) and thickness, rather than the layer resistivity or





thickness alone. Therefore, a thinner, more conductive layer (e.g., thin hypersaline lake) can produce a similar EM signal to a thicker, less conductive layer (e.g., thick package of saturated sediments) making it difficult to determine the exact layer thickness and resistivity. By integrating multiple geophysical techniques with different resolution capabilities, e.g., RES,

seismic and EM, layer thicknesses can be accurately constrained allowing reliable determination of the subsurface resistivity structure (Killingbeck et al., 2021).

The RES inferred presence of a subglacial lake beneath DIC, motivated a new campaign of multi-technique surface geophysical surveys in 2022 to examine the properties of the hypothesized lake, characterize the lake complex, and thereafter to sample the subglacial water. Active source seismic, TEM and MT data all collected in the same field campaign. The data

were recorded on two profiles and included 9 km of active source seismic reflection data, 7 TEM soundings and 17 MT stations. One profile was acquired across the location of the proposed lake, extending into a region where a brine network was believed to be present (Line A). A second profile was acquired along the long axis of the proposed lake (Line B) (Fig. 1b). In this study, we present the results from the seismic and electromagnetic data acquired over the inferred subglacial lake, which leads to a new interpretation of the subglacial material beneath DIC.

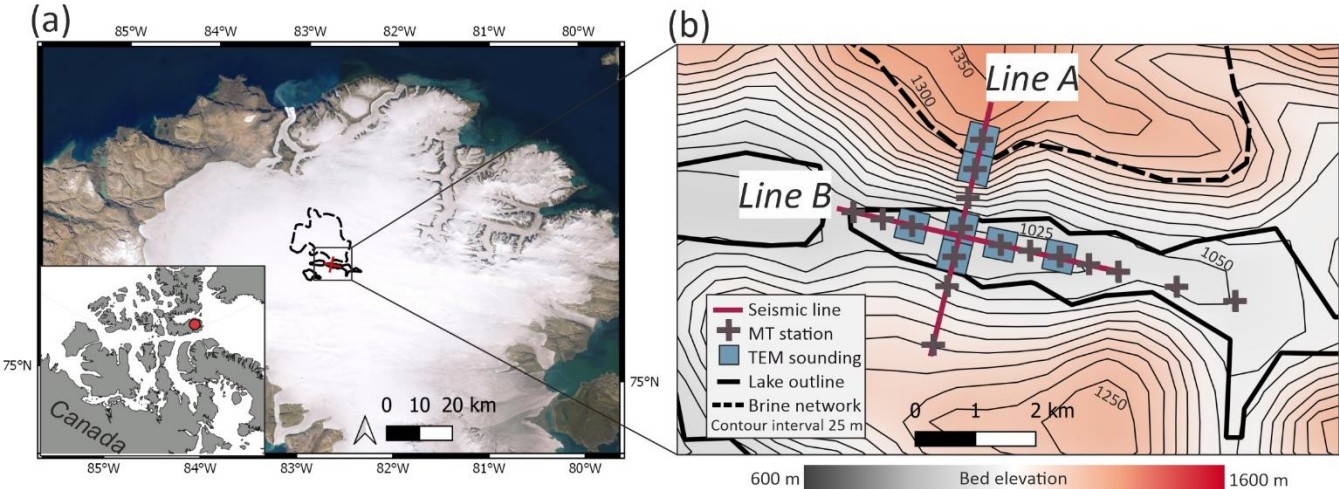

**Figure 1. (a) Regional map of DIC (ESRI satellite world imagery) with subglacial lake, T2 (black line) and brine network (black dashed line) proposed from RES (Rutishauser et al., 2018; Rutishauser et al., 2022). (b) Enlarged insert shows the bed elevation measured from RES with seismic, TEM and MT survey locations.**

## 2 Methods

### 2.1 Active-source seismic reflection

#### 2.1.1 Acquisition and processing

The active source seismic reflection data acquisition consisted of a moving spread of 48 40 Hz vertical component geophones spaced 10 m apart, along two profiles (A and B) (Fig. 1). For each spread, we collected data at 11 shot locations using an 8 kg sledgehammer impacting a thick steel plate. At each source location, at least 5 hammer shots were stacked into a single shot





gather to increase the signal to noise ratio. The shot locations for each spread were at offsets: -60 m, 0, 120 m, 240 m, 360 m,
     480 m, 530 m, 590 m, 650 m, 710 m, and 770 m from the first geophone. After data were collected at each of the 11 shot
     locations, the spread was moved 470 m along the survey line and the data collection was repeated. For line A, the seismic line
     was moved a total of 9 times, obtaining reflection points at the ice base spaced every 5 m along a 4500 m distance. For line B,
     the seismic line was moved a total of 10 times, obtaining reflection points at the ice base spaced every 5 m along a 5000 m
distance. The in-line resolution of the migrated data and the vertical resolution is 9 m (assuming ¼ wavelength, λ). In theory,
     water layers down to λ/32 (1 m) can be detected; however, amplitudes from these layers may not be representative of their
     elastic properties due to seismic tuning (Booth et al., 2012).

     Seismic processing was completed using MATLAB and the open source CREWES package available from
     www.crewes.org. The processing steps included:

1.   Apply shot-to-shot energy variation correction along the line, method follows that applied in (King et al., 2008),

         2.   Remove bad traces,

         3.   Bandpass filtering (80 Hz, 90 Hz, 250 Hz, 260 Hz),

         4.   Bandpass filtering in the FK domain (-0.025, -0.035, 0.025 0.035),

         5.   Mute airwave using a velocity of 333 m/s,

6.   Apply a top mute,

         7.   Apply FK fan filter between 500 and 5000 m/s,

         8.   Apply NMO correction using a velocity of 3700 m/s and DMO correction using a velocity of 3902 m/s over the
              steeply dipping valley side (where the dip angle is estimated at 18.510 from the Bed DEM),

         9.   Stack the data

10.  Apply post stack finite difference migration using a velocity of 3700 m/s.

### 2.1.2 Normal incident reflection method

The strength of the reflection from the base ice interface (R1) can indicate its acoustic properties and hence, allow
determination of the material that is likely present, i.e. water or rock. To estimate the acoustic properties of the ice-bed interface
(R1) we first calculate the reflection coefficient by analysing the amplitudes of R1 and its multiple. The basal reflection

coefficient $c_R$ can be determined as a function of incidence angle $\theta$ using

$$c_R(\theta) = 2 \frac{A_{M1(\theta)}}{A_{R1(\theta)}} e^{aL(\theta)} \qquad (1)$$

where $A_{R1}$ and $A_{M1}$ are the amplitude of the first and second (the multiple) ice bottom reflections, respectively, $a$ is the
absorption coefficient, and $L$ is the raypath length of the $R1$ reflection. We use the multiple bounce method (Maguire et al.,
2021; Holland et al., 2009; Horgan et al., 2021) and the normal incidence approximation where only traces with an incident

angle < 10° were used. Here, our hammer and plate impulse source has a minimum phase source signature. Therefore, our data
     is minimum phase (as we have not applied deconvolution, to zero phase our data, during the seismic processing), hence the



reflections are represented by a minimum phase wavelet. We picked the absolute maximum energy of the wavelets *R1* and *M1*, by defining a window around the minimum phase wavelet. We assumed an attenuation $a = 0.27$ km$^{-1}$ (Horgan et al., 2012). This attenuation corresponds to a seismic quality factor ($Q$) of 30–300 for 10–100 Hz waves in a 3860ms$^{-1}$ medium. We estimate $c_R$ in the trough to be $0.468 \pm 0.116$. $c_R$ can then be used to determine the acoustic impedance ($Z_b$) of *R1* using

$$Z_b = Z_{ice} \frac{1+c_R}{1-c_R} \tag{2}$$

where $Z_{ice}$ is ~ 3.33 x 10$^6$ kg.m$^{-2}$ s$^{-1}$.

## 2.2 Electromagnetics

### 2.2.1 Transient Electromagnetics

TEM data were acquired with a Geonics PROTEM67 system consisting of a three-channel digital time-domain receiver unit, a vertical component multi-turn receiver coil (area 100 m$^2$), and a TEM67 generator-powered transmitter. A 500 m x 500 m square transmitter loop was set out using snow mobiles and PVC stakes at each of the four corners, with 3.3 $\Omega$/km resistance wire. The receiver coil was placed 250 m outside of the loop, 250 m away from the generator powered transmitter module. The transmitter loop locations are shown in Fig. 1. Background noise levels, measured with the transmitter coil turned off, are considered low at DIC since there are no large sources of electrical noise, e.g., power lines, buildings, roads, metal infrastructure. Background noise readings were acquired, with noise level at 2 x 10-11 Vm$^{-2}$. At each location the transmitter module was used to power 23 A of current around the large loop (500 m x 500 m). Base frequencies of 7.5 Hz and 3 Hz were acquired with 30 measurement time gates, 120 second integration time and 10 stacks, meaning each sounding took 1 hour and 30 minutes. The time–amplitude decay curves measured during each sounding, were inverted to obtain a 1D resistivity profile with depth (Killingbeck et al., 2020).

For each sounding only the positive decays were used in the inversion process, which assumes that the subsurface acts purely as a resistor. In certain cases, the subsurface may behave like both a resistor and a capacitor known as induced polarization (IP) (Grombacher et al., 2021), however we do not invert for IP effects in this study. With this in mind, we remove all negative data points and any positive data points below the background noise level. We use the open-source MATLAB code MuLTI-TEM (Killingbeck et al., 2020) to invert the data using a trans-dimensional Bayesian inversion method that determines the posterior probability density function of resistivity as a function of depth (Killingbeck et al., 2020). The inversion is constrained by the ice thickness at each sounding location, derived from RES data (Rutishauser et al., 2022), where the resistivity bounds are limited between 10$^3$ $\Omega$.m and 10$^6$ $\Omega$.m in the ice layer. At depths below the ice, the resistivity bounds are set between 10$^{-1}$ $\Omega$.m and 10$^6$ $\Omega$.m. The inversion parameters used in MuLTI-TEM are shown in Appendix B in Table B1 and B2.



### 2.2.2 Magnetotellurics

MT uses natural electromagnetic signals to image subsurface resistivity structure. In MT exploration, the depth of investigation increases as the frequency decreases, thus, frequency can be considered a proxy for depth. Data were recorded with Phoenix Geophysics MTU-5C instruments, with serial numbers 10379 and 10351. Electric fields were measured with 100 m long dipoles connected to the ice with titanium sheet electrodes and custom high impedance amplifiers. Magnetic fields were measured with Phoenix Geophysics MTC80H induction coils that were buried in the snow. The station deployment is shown in Fig. 1. Line A had 7 stations with a 500 m spacing and was orthogonal to the trend of the trough that was inferred to contain a subglacial lake. Line B had 10 stations and was parallel to the trend of the trough. At each station MT time-series data were recorded for 24-48 hours at sample rates of 24000 Hz and 150 Hz. The stations were deployed in a geographic coordinate system because it was not possible to use a compass owing to the high magnetic inclination.

The time series data were processed using a statistically robust algorithm of Egbert (1997). This produced high quality estimates of apparent resistivity, phase, and tipper in the frequency band 100 – 0.01 Hz. The dimensionality of the DIC MT data were investigated using the phase tensor approach (Caldwell et al., 2004). The phase tensor gives a graphical representation of how the measured MT data varies with the azimuth of the coordinate system used to plot the data. This analysis of dimensionality of the data is required to determine which inversion approach (1D, 2D or 3D) is most suitable for the data.

- If the subsurface has a 1D resistivity structure, then the phase tensor will plot as a circle and the skew angle will be zero.
- A 2D resistivity structure will result in elliptical phase tensors, also with zero skew angle. The strike direction will be aligned with either the major or minor axis of the ellipse.
- A 3D structure will result in an elliptical phase tensor and a non-zero skew angle. By looking at the phase tensors as a function of frequency, information can be obtained about the depth variation of the resistivity structure.

Thereafter, a 2D inversion was implemented using the code of Rodi and Mackie (2001). The starting model was generated in the Winglink software package and included the ice layer with the base elevation taken from recent RES data (Rutishauser et al., 2022). The ice was assigned a resistivity of 100,000 $\Omega$m and fixed in the inversion. Furthermore, a tear (discontinuity in resistivity) was allowed at the base of the ice to avoid excessive smoothing. This tear enables a sharp contrast in resistivity between the ice and subglacial material directly beneath the ice in the inversion model, if required to fit the data observations. Furthermore, a number of 2D inversions were ran to allow the optimal degree of smoothing, known as the trade-off parameter ($\tau$), to be determined. Here, a value of $\tau = 3.2$ was chosen as it defines the corner of the trade-off curve representing an optimum balance between fitting the measured MT data without obtaining an unrealistically rough model (Fig. C1 in Appendix C).



## 3 Results

### 3.1 Seismic

The seismic reflection data clearly images the ice base reflector (R1) with a second reflector (R2) directly below in the trough
and southern section (Fig. 2). In the northern part of Line A, a relatively flat plateau is observed with just one primary reflector
(R1). Continuing south along line A, the steeply dipping valley side is imaged down to the trough where the lake, T2, was
thought to exist. In the trough and southern section R1 and R2 are observed and clearly separated (maximum separation ~
0.013 seconds). The long axis profile (Line B) clearly images R1 and R2 with constant separation along the trough (~ 0.011
seconds).

Here, we define the polarity of the first arrival of the direct wave as positive, identified by a negative minimum phase
wavelet (Fig 3a). With this in mind, we observe a positive polarity for the ice base interface (R1) and second reflector (R2)
(Fig. 3a), opposite to that expected for subglacial water (Fig. 3b), indicating the material directly under the ice is unlikely to
be a lake. Furthermore, if a lake existed in this bedrock trough, the polarity should change at the ice-water and ice-bedrock
interfaces along Line A, but no polarity reversal is observed between R1 and R2 (Fig. 2 and 3). Using the lake boundaries
derived from the RES analysis, we would expect R1 to onlap onto R2 at the lake edge, in the southern part of Line A. However,
in the southern section of Line A, R1 and R2 are clearly separated, with R2 continuing along the southern flank and extending
the entire length of Line A (Fig. 2).

To confirm our polarity analysis, synthetic seismograms were computed using the CREWES finite difference
algorithm (Margrave and Lamoureux, 2019) for two models of (1) a 10 m-thick lake underlying 760 m of glacial ice, and (2)
a 20 m-thick consolidated sediment package underlying 760 m of glacial ice (Fig. 3b-c; Appendix A) and compared to the
acquired shot gather (Fig. 3a). The source wavelet used in the simulations was a negative minimum phase wavelet with
dominant frequency 100 Hz, which best represents our impulse source (hammer and plate) and the direct wave observed in
our seismic data. Here, the model of a 20 m thick consolidated sediment package best matches the acquired shot gather (Fig.
3c).

Additionally, we conducted a sensitivity test for the polarity expected at an ice-water interface for different water
types: 1) water, 2) seawater, 3) brine at 0°, 4) brine at -4° and 5) brine at -10° (Fig. 4) (Booth et al., 2012; Brown, 2016; Prasad
and Dvorkin, 2004). Our sensitivity testing shows there is always a negative polarity reflection for an ice-water interface for
the scenarios tested (Fig. 4). For the ice-brine interfaces, the modelled reflection coefficients ranged from -0.4 for brine at 0°
to -0.2 for brine at -10°. Therefore, the positive polarity of R1 is also opposite to that expected for an ice-brine interface.



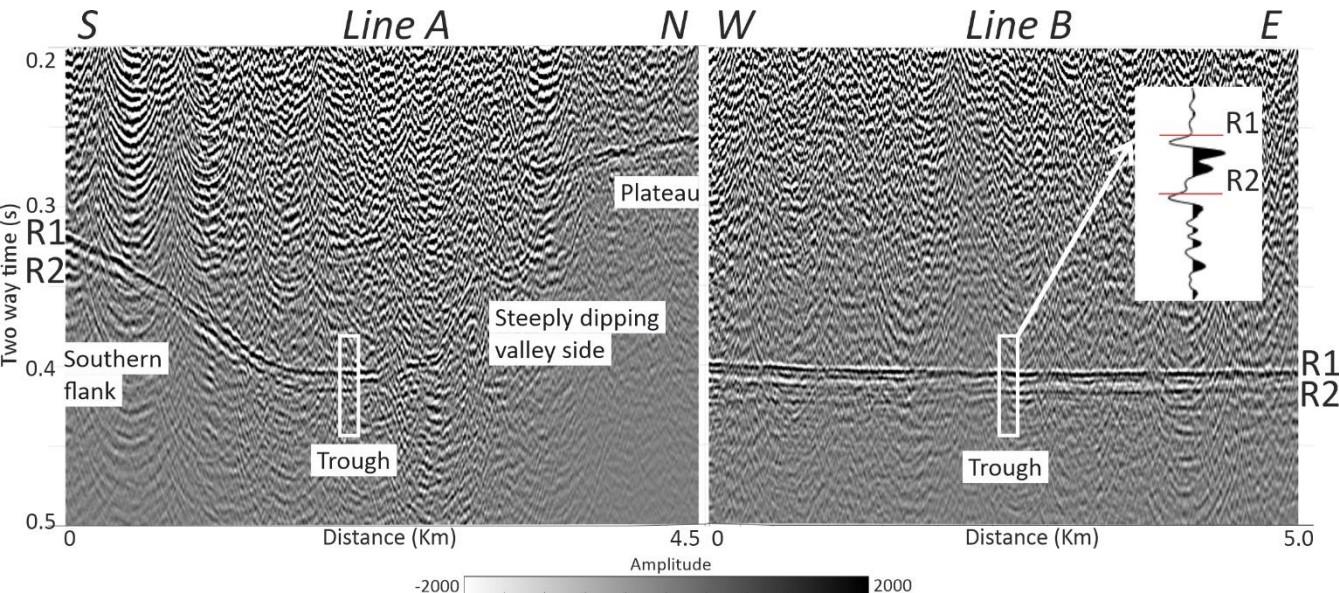


**Figure 2. Migrated seismic reflection sections for Line A and B, where the ice base reflector (R1) is highlighted, and a second reflector, R2, appears directly below in the trough and southern section. Enlarged insert showing the seismic signature of R1 and R2.**

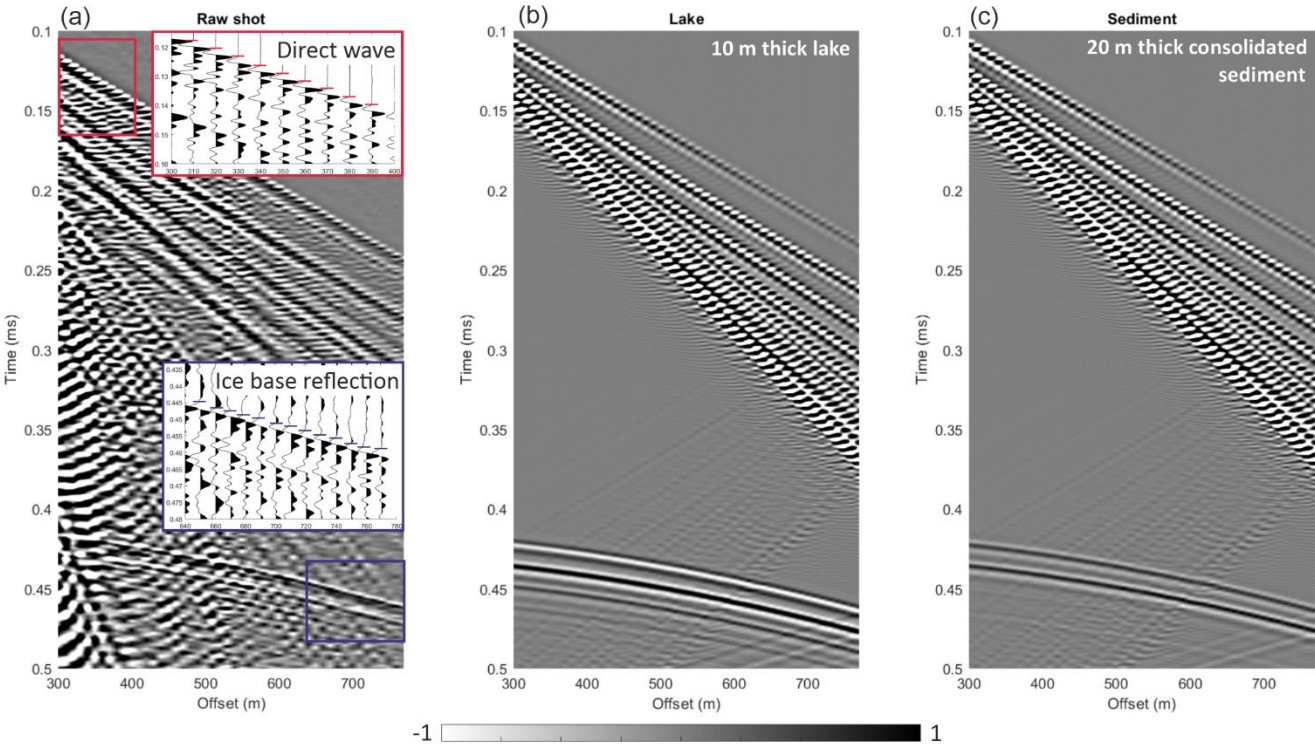

**Figure 3. (a) Raw shot gather acquired in the middle of the trough where Line A and B cross, thought to be the deepest part of the proposed lake, with zoomed inserts of the direct wave and ice base reflection. (b) Synthetic shot gather of a 10 m thick lake underlying**



**760 m of glacial ice. (c) Synthetic shot gather of a 20 m thick consolidated sediment package underlying 760 m of glacial ice. The velocity and density models used for these synthetics are shown in Appendix A (Fig. A1).**

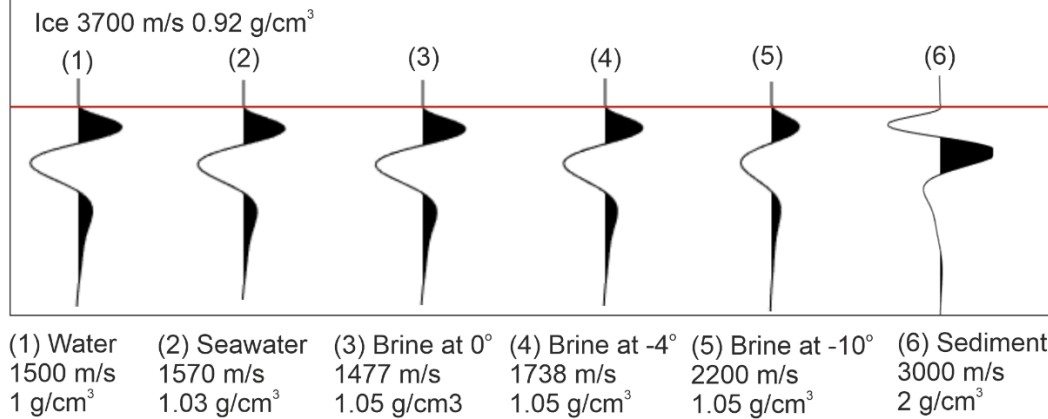

**Figure 4. Sensitivity testing for different water types: 1) water (Booth et al., 2012), 2) seawater (Brown, 2016), 3) brine at 0°, 4) brine at -4°, 5) brine at -10° and 6) consolidated sediment (Peters et al., 2008). The acoustic properties of the brine at different temperatures have come from the results of an acoustic pulse transmission experiment conducted in Prasad and Dvorkin (2004).**

### 3.2 Electromagnetics

The electromagnetic data provides independent observations that support findings from the seismic analysis and show no

evidence for a subglacial lake beneath DIC. TEM and MT are complementary methods, in which both techniques independently measure the subglacial electrical resistivity structure. Therefore, before inverting the EM data, a joint comparison of the TEM and MT measured data enabled each dataset to be evaluated for consistency. The TEM and MT observed data have very high apparent resistivity curves (Fig. 5a). The observed EM data was compared with synthetic models of 1) a hypersaline lake, 2) saturated sediments, and 3) a resistive subsurface. Here, the observed data has no resemblance to

the predictions of the 1D resistivity models representing a hypersaline subglacial lake or saturated sediments. The observed data best fits the model with a very resistive subsurface ($> 1000$ Ω.m) (Fig. 5).





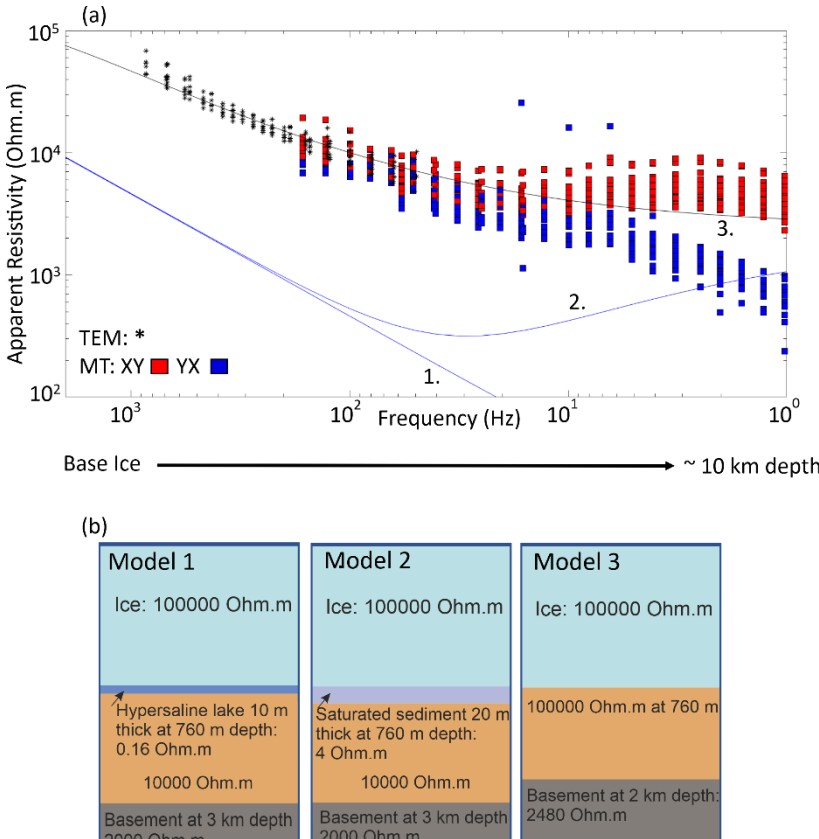

**Figure 5. (a) Frequency verses apparent resistivity of the observed TEM and MT data acquired across the whole survey area with synthetic models 1-3 plotted. (b) Synthetic models 1 – 3 resistivity structure. The time (t) after turn-off for the TEM data has been transformed to period (T) according to the transformation T = t/0.2, which has been converted to frequency (f) for the purpose of this plot by f = 1/T. For the MT data, the XY data denotes apparent resistivity and phase calculated from the North-South electric field and the East-West magnetic field. The YX data denotes apparent resistivity and phase calculated from the East-West electric field and the North-South magnetic field.**

### 3.2.1 Transient Electromagnetics

The results from the 1D MuLTI-TEM inversion for each TEM sounding acquired along Line A are shown in Fig. 6. The inverted resistivity profiles of the TEM data show highly resistive rock layers (1000 to 100000 Ω.m) directly under the ice in the trough and on the plateau with a ~ 1000 Ω.m layer at depths > 2 km. Furthermore, all inverted resistivity soundings for Line A and B (Appendix B, Fig. B1) show very similar results consisting of a highly resistive subsurface. Of note, given a choice between a simple model (e.g. thick layers with small resistivity changes) and complex model (e.g. thin layers with large resistivity changes) which provides a similar fit to the data, the Bayesian inversion method will always choose the simple model. Additionally, the TEM method can resolve conductive structures more accurately than resistive ones, highlighted by a tighter probability density function over the lower ~ 1000 Ω.m layer compared to the resistive upper layer shown in Fig. 6 and Fig. B1.





With these points above in mind, we tested the vertical resolution of the TEM technique to determine how thin a
conductive layer, ranging from a hypersaline lake to saturated sediments, could be imaged in comparison to the TEM data
acquired on DIC. Multiple synthetic models were created using the forward modelling code in MuLTI-TEM. A 3 layer model
was used with 1) a 100000 Ω.m layer overlaying 2) a layer with variable thickness and resistivity with 3) a 10000 Ω.m
basement. The different resistivities tested for layer 2 were 0.1 Ω.m (representing a hypersaline lake), 1 Ω..m, 10 Ω.m and
10000 Ω.m. The different thicknesses tested for layer 2 were 0.01 m, 0.1 m, 1 m, 10 m and 100 m. Figure 7 shows the results
from this test highlighting a 0.01 m thick layer of 0.1 Ω.m is resolvable compared to the resistive subsurface observed beneath
DIC. Also, a 0.1 m, 1 Ω.m layer and a 1 m thick 10 Ω.m layer is resolvable compared to the resistive subsurface observed
beneath DIC. These results are highlighted by the separation of the green, blue, and brown curves from the resistive model in
black, providing further evidence for the lack of subglacial water directly beneath DIC.

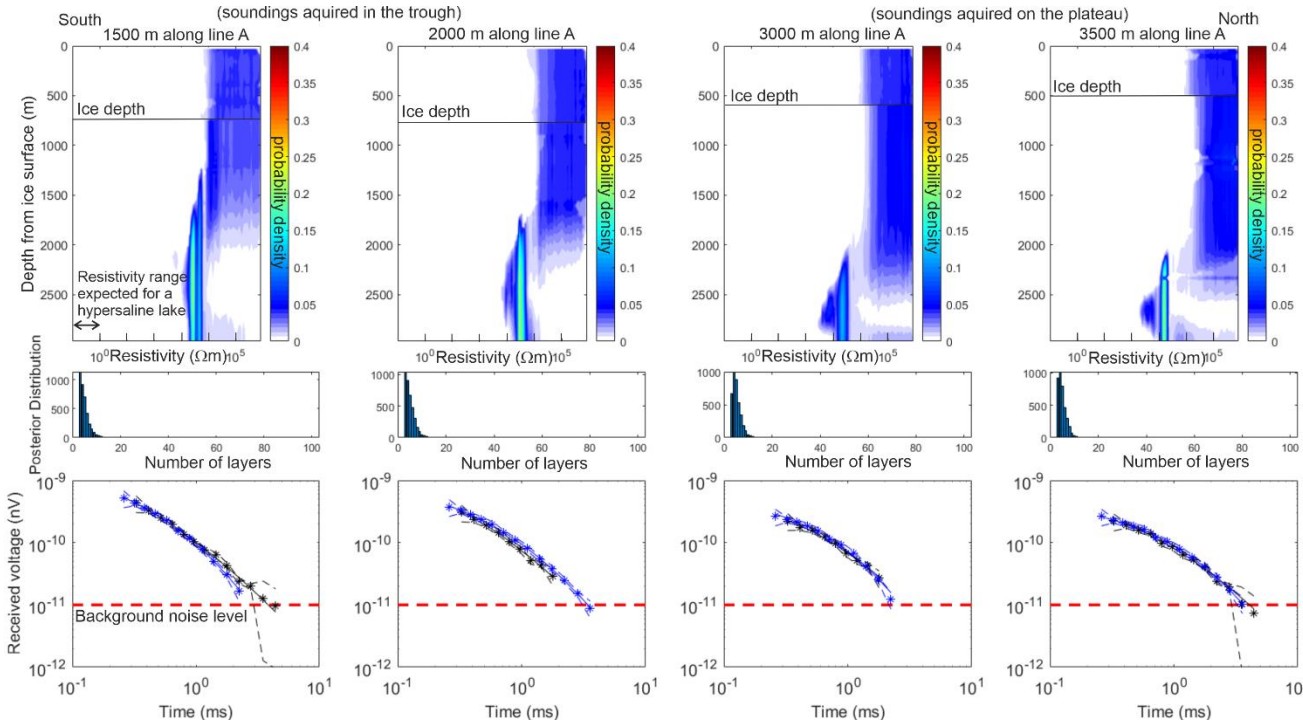

**Figure 6. Line A TEM Bayesian inversion results for the 7.5 Hz and 3 Hz base frequencies. Top plot: posterior distribution of
resistivity with depth. Middle plot: posterior distribution of the number of layers. Bottom plot: comparison of data fit plot, with the
best fitting forward model accepted in the ensemble (-) compared to the data (*) and error tolerance (--). The black data points are
the 3 Hz base frequency, and the blue data points are the 7.5 Hz base frequency. The red dashed line is the background noise level.**





**Figure 7. TEM vertical resolution analysis of synthetic 3 layered models compared to the DIC TEM data. Synthetic models are shown by the green, blue, brown and black lines which include: a 100000 Ω.m layer, overlaying a layer with variable thickness and resistivity (0.1 Ω.m, 1 Ω..m, 10 Ω.m and 10000 Ω.m, respectively) with a 10000 Ω.m basement. The different thicknesses for layer 2 are: (a) 0.01 m, (b) 0.1 m, (c) 1 m, (d) 10 m and (e) 100 m. The DIC TEM data is shown in grey with the background noise levels marked by the red dashed line.**

### 3.2.2 Magnetotellurics

Analysis of the MT apparent resistivity curves (Fig. 5a) shows that at the highest frequencies (100 - 30 Hz) the XY and YX apparent resistivity are similar, suggesting the subsurface resistivity structure may be relatively 1D. Here, the phase tensors show relatively high skew angles, indicating complicated near surface resistivity structure (Fig. 8a). Below a frequency of 30



Hz the XY and YX apparent resistivity curves separate, indicated a resistivity structure that is 2D or 3D (Fig. 5a). Here, the
phase tensors are aligned with the major axes perpendicular to the trend of the subglacial trough, and Line B. While the skew
angles are not zero, the data in the frequency band 100 – 1 Hz show evidence for a 2D behaviour with a strike of N105°E (Fig.
8b-c). The data were rotated to a co-ordinate system with a strike direction of N105°E. The XY data were defined as the TE
mode and the YX data defined as the TM mode. The measured data show that the TE mode has much lower apparent resistivity
values at low frequency than in the TM mode (Fig. C2).

Three 2D inversions were undertaken for Line A. Line B cannot be inverted with a 2D approach since it is parallel to
the geoelectric strike. The first inversion ran for Line A began from the model with the ice layer. Error floors of 10% and 5%
were applied to the apparent resistivity and phase respectively. Data in the frequency band 100 – 1 Hz were selected for
inversion to focus on the shallow structure. The $\tau = 3.2$ inversion reached an r.m.s. misfit of 1.356 after 200 iterations. The fit
is shown in Fig. C2 and good agreement between the measured and predicted MT data can be seen. The final resistivity model
is shown in Fig. 9a where the subglacial resistivity is very high, 3000 to 10,000 Ωm; whereas hypersaline water is expected to
have a resistivity of ~0.16 Ω.m (Killingbeck et al., 2021). Two features with lower resistivity can be observed in the model.
The first is a layer with resistivity in the range 300 – 1000 Ωm located at a depth of 3-4 km below the surface and dipping to
the north. This feature is responsible for the decreasing apparent resistivity as a function of frequency at all stations. The second
is a more subtle ~ 3000 Ωm layer located directly beneath the deepest part of the trough. With the limited number of MT
stations this resistivity feature is only resolved by 1 or 2 MT stations. Two additional inversions were performed to determine
if this feature is required by the data.

The pseudo section in Fig. C2 shows evidence for a static shift in the TM apparent resistivity at the station distanced
1 km along Line A. A static shift is a frequency independent offset in apparent resistivity and is often caused by near surface
heterogeneity. The second inversion for Line A allows static shifts to be present in both the TE and TM mode data. The second
295    inversion ran for Line A produced a model that was similar to that obtained in the first inversion (Fig. 9b).

To determine if the relatively low resistivity (~ 3000 Ωm) basal layer was present beneath the ice, a final inversion
was performed with a second tear permitted at a depth of 100 m beneath the base of the ice. This third inversion showed that
a ~ 3000 Ωm layer was consistent with the data, but not required (Fig. 9c). The lack of high frequency MT data limits the
resolution of the shallowest subglacial structure that can be resolved. However, these higher frequencies were obtained from
300    the TEM data and a joint comparison of the TEM and MT measured data with synthetic models showed the observed EM data
is very different to that expected for a subglacial hypersaline lake (Fig. 5).



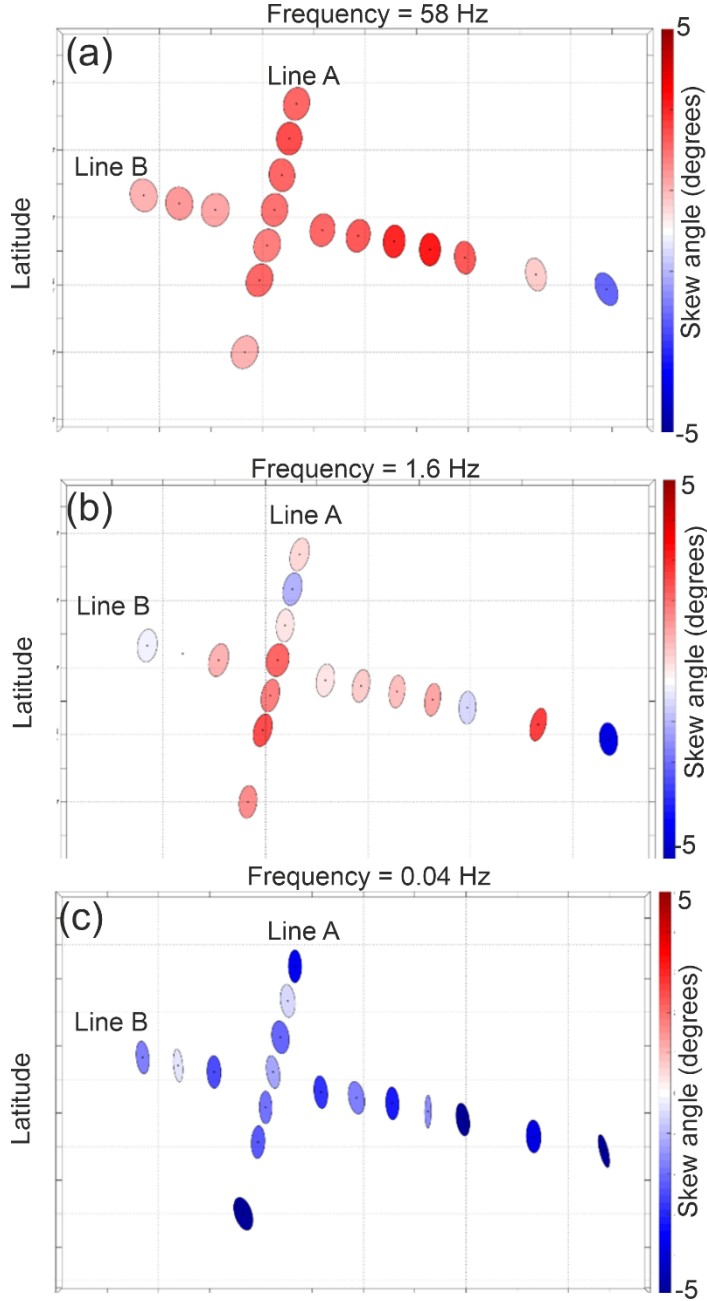

**Figure 8. Phase tensor at frequencies of 58 Hz, 1.6 Hz and 0.04 Hz plotted in map view. The direction of the major and minor axes of the ellipses show to possible strike directions. Circles indicate a 1D structure while ellipses indicate a 2D or 3D structure. This direction variation varies with frequency but is consistent with the direction of the trough axis which is N105E. The colour fill shows the skew angle of (Caldwell et al., 2004). Values close to zero indicate a 1D or 2D resistivity structure. Non-zero values indicate a 3D resistivity structure.**





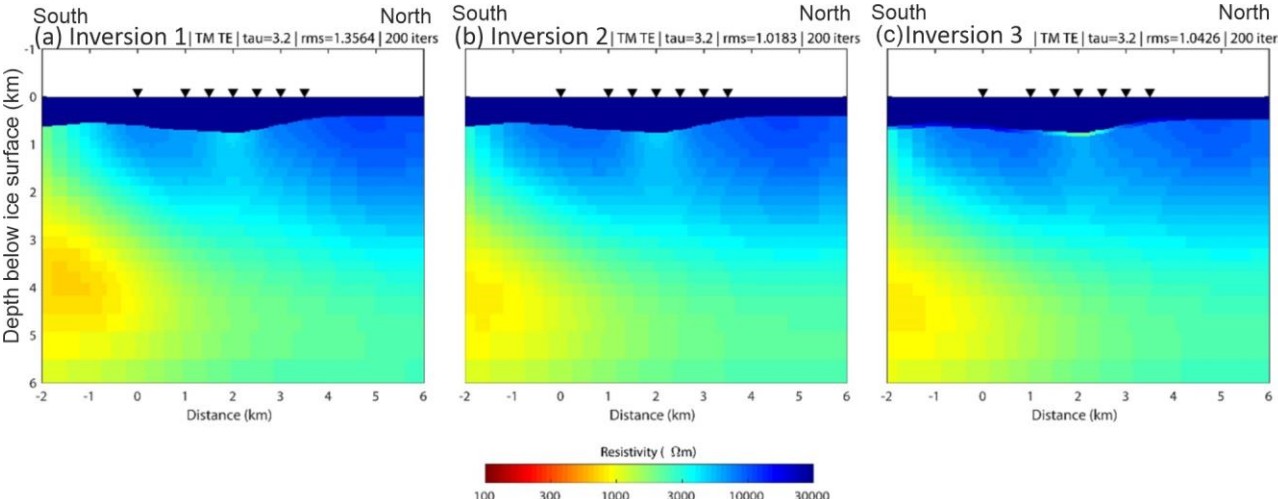

**Figure 9. 2D models for Line A obtained by inversion with the algorithm of Rodi et al. (2001). The starting model included the ice layer with a resistivity of 100,000 Ωm. (a) Inversion 1. (b) Inversion 2 with static shifts free at the station 1 km along the line. (c) Inversion 3 with statics shifts free at the station 1 km along the line and a second tear added 100 m below the base of the ice. Trade-off parameter for all three inversions was τ = 3.2.**

### 3.3 Properties of the material directly under DIC

The acoustic impedance of the material in the trough, directly beneath the centre of DIC, is estimated using the normal incident reflection coefficient method. Here, the estimated acoustic impedance of reflector R1 is $9.49 \pm 1.92 \times 10^6$ kg m$^{-2}$ s$^{-1}$, comparable to the hardest rocks and frozen sediments at Subglacial Lake Ellsworth, Antarctica (Smith et al., 2018) and significantly higher than that of water $1.5 \times 10^6$ kg m$^{-2}$ s$^{-1}$ (Fig. 10a). Laboratory studies which measured the acoustic properties of permafrost samples collected in the Canadian Arctic, show that an increase in clay content results in a decrease in seismic velocity at temperatures < 0°C (King, 1984). An additional study, measuring the effect of porosity and pore fluid salinity on seismic velocity has shown an increase in porosity and pore fluid salinity can lead to a decrease in the seismic velocity at temperatures < 0°C (Pandit & King, 1979). Direct comparison of these studies to the estimated acoustic impedance of the subglacial material in the trough beneath DIC, suggest the material could have clay content < 40% and a low content of unfrozen saline pore fluid (Fig. 10a).

The inverted resistivity profiles of both the TEM and MT measurements suggest highly resistive rock layers (1000 – 100000 Ω.m) are present directly beneath the ice, with a 1000 Ω.m layer at depths > 2 km (Fig. 6, 9 and B1). These resistivity values are ~3 orders of magnitude greater than those expected in the presence of a hypersaline lake (~0.16 Ω.m; Killingbeck et al., 2022), a freshwater lake (1 - 10 Ω.m; Christner et al, 2014; Priscu et al., 2021) or unfrozen saturated sediment (~1 - 100 Ω.m; Gustafson et al., 2022) (Fig. 10b). Laboratory studies which measured the resistivity of Canadian Arctic permafrost and sedimentary rocks at temperatures < -10°C show that a high clay content and saline pore fluid can decrease the resistivity by orders of magnitude (Pandit & King, 1979; King et al., 1988). Direct comparison of these studies to the estimated resistivity



of the subglacial material in the trough beneath DIC, supports our interpretation from the acoustic impedance analysis, that the material is likely to have a low clay content and low content of unfrozen saline pore fluid.

**Figure 10. Properties of the material in the trough directly beneath DIC. a) Acoustic impedance compared to that of subglacial material from subglacial lake Ellsworth: 1) lakebed, subaqueous, soft wet sediment, (2) subglacial, soft wet sediments, (3) hard bed, wet sediments and (4) hardest bed, rock or frozen sediment (Smith et al., 2018), and laboratory studies measuring the acoustic properties of permafrost and sedimentary samples collect in the Canadian Arctic at -15°C (King, 1984; Pandit & King, 1979). Horizontal coloured lines represent estimated acoustic impedance values for different materials detailed in Peters et al. (2008). b) Resistivity compared to that of subglacial lake Whillans (SWL; Christner et al, 2014) and Mercer (SLM; Priscu et al., 2021) direct samples, airborne electromagnetic (AEM) measurements from the Taylor Dry Valleys, Antarctica ((5) Blood Falls outflow, (6) west lake Bonney between 5 m and 35 m depth, (7) lake Fryxell between 5 m and 18 m depth, (8) sediments with brine in the pores and**



**(9) glacier ice (Mikucki et al., 2015), and laboratory studies measuring the resistivity of permafrost and sedimentary samples collect in the Canadian Arctic at temperature <-10°C (Pandit & King, 1979; King et al., 1988). Horizontal coloured lines represent estimated**
**resistivity values for different materials detailed in Killingbeck at al., (2021) and Key and Siegfried (2017).**

## 4 Re-evaluation of the airborne reflectivity data

With these new seismic and electromagnetic data observations in mind, we reanalysed the RES data which lead to the original inference of the presence of a subglacial lake beneath DIC (Rutishauser et al., 2018; Rutishauser et al., 2022). In the RES studies (Rutishauser et al., 2018; Rutishauser et al., 2022), the identification of a subglacial lake was made based on relatively

high basal RES reflectivity and specularity content, located in a hydraulically flat region. Here, we explore an alternative analysis of the RES data which fits the seismic and electromagnetic observations of dry or frozen, non-conductive basal material.

To calculate basal reflectivity, the radar energy loss through dielectric absorption of the overlying ice (englacial attenuation rates) must be estimated. Attenuation rates are commonly derived directly from RES data via linear regression fits

between the observed bed power and ice thickness (Rutishauser et al., 2018; Rutishauser et al., 2022; Gades et al., 2000; Schroeder et al., 2016; Schroeder et al., 2016) (Fig. 11a). However, attenuation rates can also be predicted from a temperature- and chemistry-dependent Arrhenius equation (MacGregor et al., 2007; MacGregor et al., 2015). At DIC, the linear regression fit method from the previous studies provided attenuation rate estimates of 21.8 dB/km (regression fit over the entire dataset; Rutishauser et al., 2022) and 26.8 dB/km (mean of a regression fit on a profile-by-profile basis; Rutishauser et al., 2018),

yielding a relatively high basal reflectivity that was interpreted as a subglacial lake (Fig. 11b). However, considering the new seismic, TEM and MT results, we hypothesize that these attenuation rates were overestimated, leading to an overestimation of the basal reflectivity. We therefore re-evaluated the attenuation rates (radar-derived vs. Arrhenius modeled) and resulting basal reflectivities.

### 4.1 Derivation of Arrhenius-modeled Attenuation Rates

Here, we derive new attenuation rates using the temperature- and chemistry-dependent Arrhenius equation. The derivation and description of Arrhenius-modeled attenuation rates closely follows a previous application over DIC (Rutishauser, 2019). Arrhenius-modeled attenuation rates are derived via the relationship between the radar attenuation rate $N_a$ and the high-frequency limit of the electrical conductivity $\sigma_\infty$ (measured in μSm$^{-1}$, Equation 3), which is related to the ice impurity concentration and temperature via the Arrhenius-type conductivity model (Equation 4):

$$N_a = \frac{10 log_{10} e}{1000 \varepsilon_0 c \sqrt{\varepsilon}} \sigma_\infty,$$ (3)

where $\varepsilon_0$ is the permittivity- and c the speed of light in vacuum, and $\varepsilon = 3.15$ is the dielectric permittivity of ice. The Arrhenius conductivity model is expressed as

(4)

$$\left[ \frac{E_{pure}}{ } \left( \frac{1}{ } \quad \frac{1}{ } \right) \right]$$



$$+\mu_{H^+}[H^+]exp\left[\frac{E_{H^+}}{k}\left(\frac{1}{T_r}-\frac{1}{T}\right)\right]$$

$$+\mu_{Cl^-}[Cl^-]exp\left[\frac{E_{Cl}}{k}\left(\frac{1}{T_r}-\frac{1}{T}\right)\right]$$

$$+\mu_{NH_4^+}[NH_4^+]exp\left[\frac{E_{NH_4^+}}{k}\left(\frac{1}{T_r}-\frac{1}{T}\right)\right],$$

where $\sigma_{pure}$ and $E_{pure}$ are the conductivity and activation energy for pure ice, respectively, $k = 1.38 \times 10^{-23} J K^{-1}$ is the Boltzmann constant, $T$ is the ice temperature and $T_r$ is a reference temperature. For the impurities H[+], Cl[-] and NH$_4$[+], $\mu_x$ is the

molar conductivity, $[x]$ is the molarity and $E_x$ is the activation energy. Values for the molar conductivities and pure ice conductivity were taken as the M07 $\sigma_\infty$ model for the Greenland Ice Sheet described in (MacGregor et al., 2015) and applied by (Jordan et al., 2016). To model attenuation rates over DIC, impurity concentrations were derived as the mean of measured concentrations along a 20 m deep firn core retrieved on DIC in 2015 (Criscitiello et al., 2021). All parameters used in the $\sigma_\infty$ model are detailed in Table D1 in Appendix D. The temperature-attenuation rate models and application to two example ice

temperature profiles are shown in Fig. D1 in Appendix D.

Ice temperatures used in the Arrhenius model are estimated from a 1D steady-state advection-diffusion model (Cuffey and Paterson, 2010), previously applied to DIC (Rutishauser et al., 2018; Rutishauser et al., 2022). This temperature model ignores horizontal temperature exchanges as well as basal frictional heating, strain heating from ice deformation and potential latent heat contributions from refreezing of percolated meltwater in the firn or water at the base of the ice. Flow regime

classifications (Burgess et al., 2005) show that the central part of DIC lies within flow regime 1 (FR1, $\frac{v}{d} \leq 0.05\ yr^{-1}$, where $v$ is the ice surface velocity and d the ice thickness) for which flow is driven by internal deformation, and thus the underlying basic assumptions for a 1D advection-diffusion model are valid.

Ice temperature profiles over DIC are calculated for each grid cell of the ice thickness data by (Rutishauser et al., 2022), using a geothermal heat flux of 65±5 mW m[-2] (Grasby et al., 2012), an accumulation rate of 0.19±0.05 m water

equivalent per year (Paterson 1976; Reeh and Paterson 1988) converted to downward velocity using a firn density of 330 kg/m[3], and a mean annual air temperature derived via scaling a reference temperature of -23±1°C at 1825 m asl. (Kinnard et al., 2006) with a 4.1°C/km lapse rate (Rutishauser et al., 2018; Gardner et al., 2009) over the entire ice cap. Finally, data points outside of FR1 ($\frac{v}{d} > 0.05\ yr^{-1}$), calculated from velocities reported in Van Wychen et al. (2014) are excluded. Uncertainties from the ice temperature and Arrhenius model are propagated, leading to attenuation rate uncertainties over DIC between 4.6-

7.5 dB/km, and a mean of 6.1 dB/km (Fig. D2 in Appendix D).

Here, the Arrhenius modelled attenuation rates over DIC range between 14-23 dB/km (Fig. 11c), with a mean of 17 dB/km along all survey lines, and 15.6 dB/km over the lake area (Fig. 11). These attenuation rates are significantly lower than the radar-derived attenuation rates and applying them to correct the returned bed power yields a much lower basal reflectivity in the trough (5.7 dB and 3.2 dB, respectively, Fig. 11d, Table 1). These lower reflectivities do not meet the basal reflectivity



threshold expected for the presence of subglacial water (Carter et al., 2007) and would have not led to the interpretation of a subglacial lake.



**Figure 11. Re-analysis of radar attenuation rates and reflectivity over DIC. (a) Linear regression fits (over entire DIC datasets) to derive attenuation rates ($N_{global}$) from the Operation Ice Bridge (OIB) MCoRDS radar data (2011, 2012) used in (Rutishauser et al., 2018) (grey) and the SRH1 HiCARS data used in (Rutishauser et al., 2022) (black). (b) SRH1 basal reflectivity (Rutishauser et al., 2022) corrected using a constant ($N_{global}$) rate of 21.8 dB/km (linear regression in a). (c) Arrhenius modeled attenuation rates over DIC. Black and grey thin lines are the SRH1 and OIB 2011/2012 survey lines, respectively. (d) SRH1 basal reflectivity corrected using the Arrhenius modelled attenuation rates ($N_{Arrh}$, shown in c). (e) Attenuation rates derived from the SRH1 dataset via an adaptive fitting approach (Schroeder et al., 2016; Chu et al., 2021). Black squares indicate areas where the minimum fit criteria are**





**not met (Appendix D). (f) OIB 2011/2012 basal reflectivity corrected using the Arrhenius modelled attenuation rates (shown in c). All attenuation rates are noted as one-way attenuation rates.**

**Table 1. Comparison of radar-derived and Arrhenius modelled attenuation rates (one-way) over DIC. Including the resulting mean basal reflectivity (R) over the previously hypothesized subglacial lake area.**

| Dataset | Attenuation rate technique | Attenuation rate (dB/km) | Basal reflectivity over subglacial lake area (dB) |
|---|---|---|---|
| SRH1 (Rutishauser et al., 2022) | Linear regression fit over all data, $N_{global}$ | 21.8 | 10.8 |
| | Adaptively fitted (mean) | 14.2-39.9 (24.6) | 8.9 |
| | Arrhenius-modeled $N_{Arrh}$ (mean) | 14.2-23 (16.9) | 5.7 |
| OIB 2011, 2012 (Rutishauser et al., 2018) | Linear regression fit over all data, $N_{global}$ | 24.3 | 10.3 |
| | Arrhenius-modeled $N_{Arrh}$ (mean) | 14-21.4 (16.8) | 3.2 |

## 4.2 Adaptive attenuation rate fitting

In addition to the Arrhenius modelling approach, we also test an adaptive bed power to ice thickness fitting approach (Schroeder et al., 2016; Chu et al., 2021) (Fig. 12). We use the SRH 1 dataset (Rutishauser et al., 2022) and derive attenuation rates at evenly distanced (2.5 km) grid points over the DIC survey area. For each point, we use an initial search radius of 5 km to calculate the correlation-coefficient magnitude between the ice thickness and attenuation corrected bed power for attenuation

rates ranging between 0-40 dB/km. Then, correlation-coefficient fit conditions are evaluated, and if not met, the search radius is extended (each round by 1 km) until a maximum radius of 25 km. We set the minimal fit criteria to 1) the minimum correlation-coefficient magnitude $C_w \leq 0.01$, 2) an initial correlation-coefficient magnitude $C_0 \leq 0.5$, and 3) the radiometric resolution $N_h \leq 3$ dB/km (Schroeder et al., 2016).

While the general distribution of increasing attenuation rates towards the margins (Fig. 12e) is reasonable, the spatial

distribution reveals abrupt transition zones that may be artefacts of the method rather than abrupt changes in the ice properties. Furthermore, the minimum fitting criteria are not met over most of the southern catchment area on DIC, including the hypothesized subglacial lake region, highlighting the difficulties in applying this method to the DIC RES dataset.





**Figure 12. Results and correlation-fit parameters from the adaptive attenuation fitting approach (Schroeder et al., 2016; Chu et al., 2021). (a) Half-width of the correlation coefficient minimum Nh. (b) Uncorrected correlation coefficient magnitude C0. (c) Minimum correlation coefficient Cm. (d) Search radius used to derive the resulting attenuation rate (either criteria are met, or maximum search radius of 25 km is reached). (e) One-way attenuation rate. Black squares mark areas where the minimum fit criteria of Nh ≤ 3 dB/km (a) or C0 ≥ 0.5 (b) are not met. (f) Relative basal reflectivity upon application of the attenuation rates in (e).**

## 5 Conclusions

In this study, we provide new geophysical evidence which shows that the proposed hypersaline subglacial lake beneath DIC is unlikely to contain water and has been misidentified. Seismic analysis shows that the ice base interface in the location of the proposed lake has a positive reflection, and the acoustic impedance of the material is estimated to be $9.49 \pm 1.92 \times 10^6$ kg m$^{-2}$ s$^{-1}$, comparable to the hardest rocks and frozen sediments at Subglacial Lake Ellsworth, Antarctica. The inverted resistivity profiles of both the TEM and MT measurements suggest highly resistive rock layers (1000 – 100000 Ω.m) directly under the ice, comparable to Arctic permafrost and bedrock. Re-evaluation of the airborne reflectivity data at DIC shows that the RES



attenuation rates (derived directly from the RES data via a linear regression fit between the observed bed power and ice thickness) were likely overestimated, leading to an overestimation of the basal reflectivity in the original RES studies. Here, we derived new attenuation rates using the temperature- and chemistry-dependent Arrhenius equation and applied them to correct the returned bed power. Our new analysis shows the bed power does not meet the basal reflectivity threshold expected

over subglacial water.

This example shows that the detection of subglacial lakes by RES is highly sensitive to the attenuation rate applied. Different methods for calculating attenuation rates may yield different results and should be rigorously assessed during RES analysis. Evidently, sensitivity studies on radar attenuation rates are a critical step in RES processing and fundamental to accurately identify subglacial lakes. It is important that future RES investigations include an attenuation-rate sensitivity

assessment so that the uncertainty and limitations of each dataset can be quantified. Furthermore, our study highlights that the acquisition of multiple geophysical techniques, where logistically possible, are essential to reliably interpret subglacial water systems.

**Appendix A: Active-source seismic refection synthetic modelling**

For our synthetic seismograms, the near surface Vp structure of the snow, firn and ice was derived from refraction observations

in the seismic data, where the travel times of the first arrivals were picked and a tomography inversion applied. This was used as the shallow Vp profile (0 - 80 m) of the synthetic models. The seismic velocity profiles for the two models are shown in the inserts in Fig. A1a and A1b.

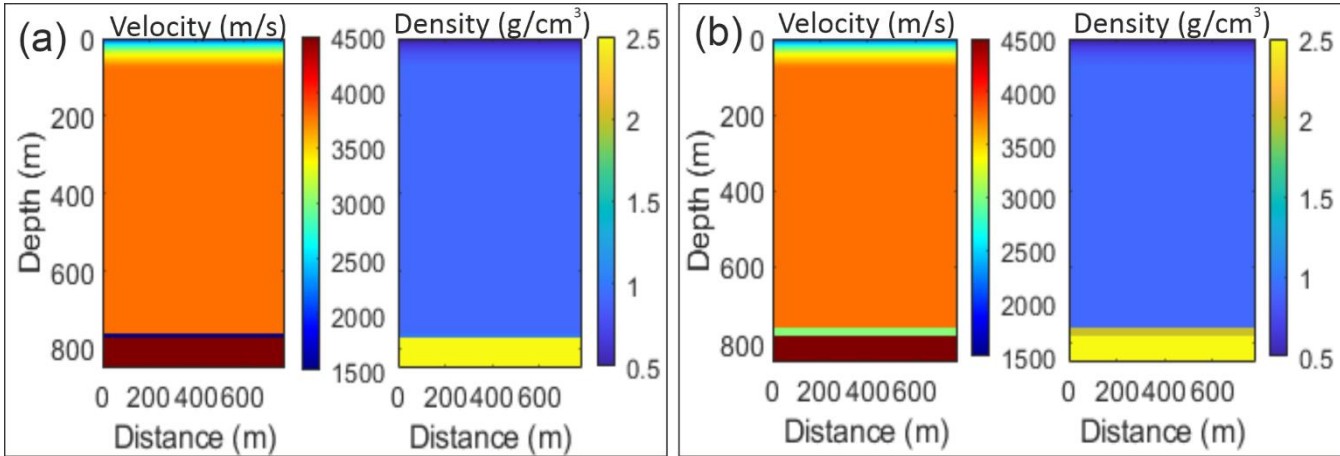

**Figure A1. (a) Vp and density model used to compute the synthetic model shown in C. (b) Vp and density model used to compute the synthetic model shown in D. (c) Devon shot gather compared with synthetic seismic data for a 10 m thick subglacial lake, with a zoomed in window of the reflected wavelets.**



## Appendix B: Transient electromagnetics

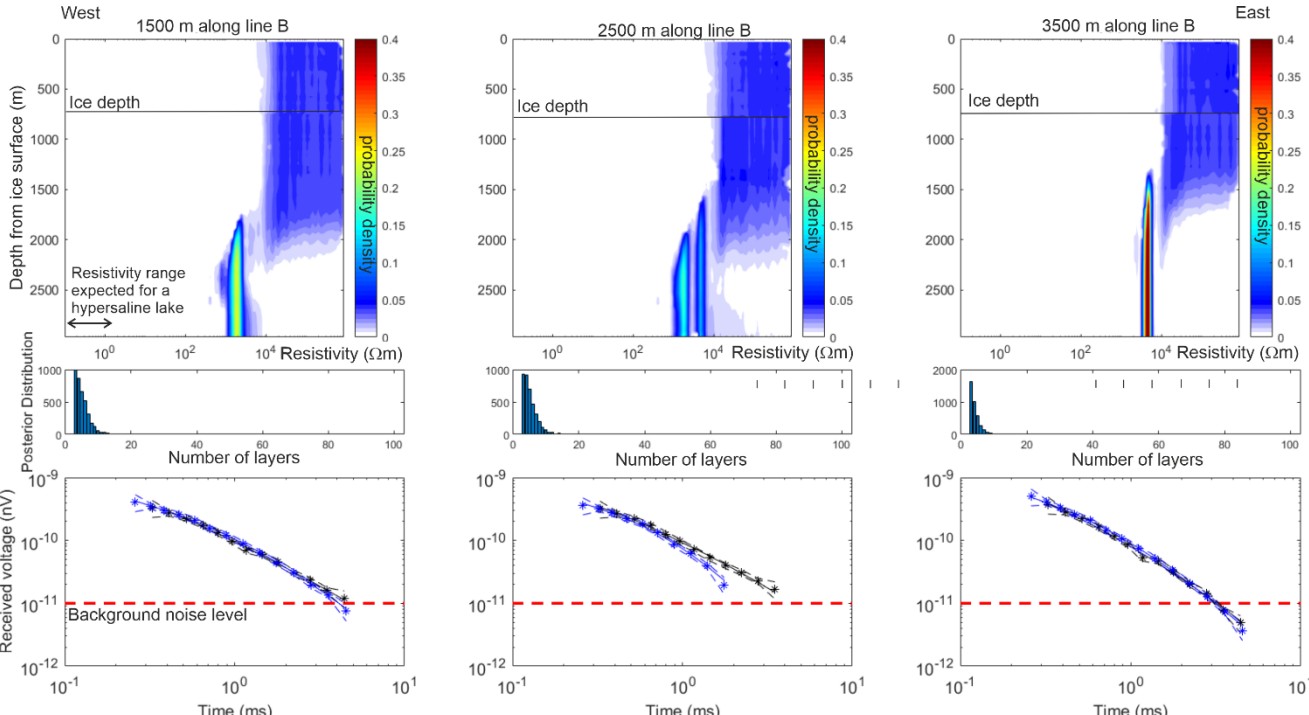

Figure B1. Line B TEM Bayesian inversion results for the 7.5 Hz and 3 Hz base frequencies. Top plot: posterior distribution of resistivity with depth. Middle plot: posterior distribution of the number of layers. Bottom plot: comparison of data fit plot, with the best fitting forward model accepted in the ensemble (-) compared to the data (*) and error tolerance (--). The black data points are the 3 Hz base frequency, and the blue data points are the 7.5 Hz base frequency. The red dashed line is the background noise level.



**Table B1. TEM survey parameters input into MuLTI-TEM (Killingbeck et al., 2020).**

| Parameter name | Unit | Parameter description | Devon Ice Cap parameters |
|---|---|---|---|
| REFTYM | msec | Time from which TOPN & TCLS are measured. For example, this could be signal off-time or start of downward ramp. | 83.25 |
| OFFTYM | msec | Time between end of one pulse and the start of the next pulse (of opposite sign) since a bipolar waveform is assumed. This is most likely equal to ¼ period of the complete waveform. For systems which have a signal which is always on, OFFTIME = 0. | 166.5 |
| TXON | msec | Digitised time of each point in the waveform (fixed at 4 points). In most cases, TXON(1) = 0, TXON(2) = pulse on-time, TXON(3) = pulse off-time, TXON(4) = REFTYM where TXON(4) - TXON(3) = turn off time. | [0.0, 0.31 82.94, 83.25] |
| TXAMP | amps | Transmitter current at time TXON(J). If signal is normalised, this should be 1. | [0.0, 1.0, 1.0, 0.0] |
| TOPN | msec | Start times of receiver windows, the number of time gates is 30. | [0.080000, 0.100000, 0.126300, 0.158800, 0.202500, 0.257500, 0.327500, 0.412500, 0.520000, 0.650000, 0.800000, 0.963000, 1.175000, 1.450000, 1.788000, 2.225000, 2.790000, 3.500000, 4.413000, 5.575000, 7.050000, 8.940000, 11.33800, 14.40000, 18.31000, 23.30000, 29.66300, 37.80000, 48.15000, 61.36000] |
| TCLS | msec | End times of receiver windows, the number of time gates is 30. | [0.100000, 0.126300, 0.158800, 0.202500, 0.257500, 0.327500, 0.412500, 0.520000, 0.650000, 0.800000, 0.963000, 1.175000, 1.450000, 1.788000, 2.225000, 2.790000, 3.500000, 4.413000, 5.575000, 7.050000, 8.940000, 11.33800, 14.40000, 18.31000, 23.30000, 29.66300, 37.80000, 48.15000, 61.36000, 78.2000] |
| SXE | meters | East coordinate of vertex I for loop position J, fixed at 4 vertices. Note the transmitter is fixed on the ground (Z=0) in this adapted Leroi code. | [250, -250, -250, 250] |
| SXN | meters | North coordinate of vertex I for loop position J, fixed at 4 vertices. | [250, 250, -250, -250] |
| RXE | meters | Receiver easting. | 500 |
| RXN | meters | Receiver northing. | 0 |
| RXZ | meters | Receiver z (always be 0 for ground-based TEM). | 0 |






**Table B2. MuLTI_TEM inversion parameters.**

| Inversion Parameter | Value |
|---|---|
| Number of Layers (constrained) | 2 |
| Weighting (data variance, σ) <br> % is of the signal at each timegate | 30% for the first 2 data points <br> 10% for the middle data points <br> 90% for the last 2 data points |
| Minimum number of total floating nuclei | 0 |
| Maximum number of total floating nuclei | 100 |
| Maximum depth | 3000 m |
| Burn in number | 10 000 |
| Number of Iterations (including burn in) | 200000 |
| Number of MCMC chains | 1 used in analysis but up to 4 are tested at each sounding to check convergence |
| Sigma resistivity change (log(R)) | 2 |
| Sigma move (meters) | 500 |
| Sigma birth (log(R)) | 2 |



**Appendix C: Magnetotellurics**

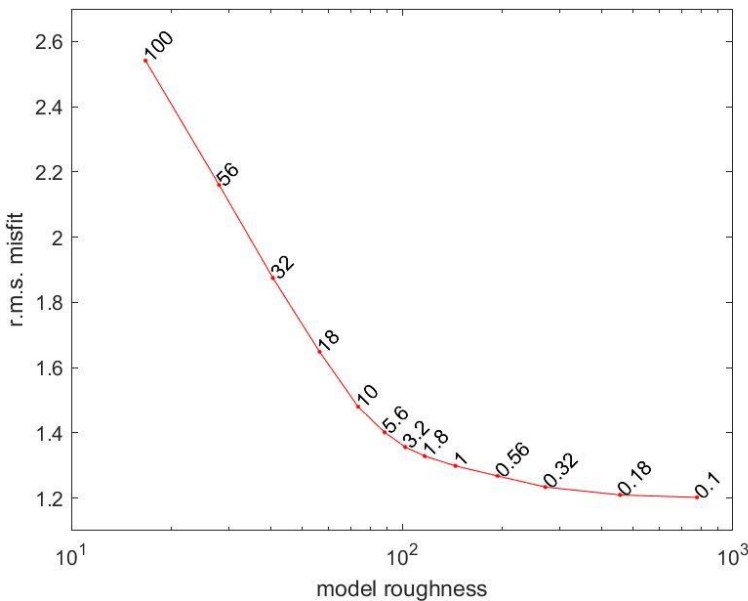

**Figure C1. L-curve for inversion 1. A value of τ = 3.2 represents a compromise between reducing the r.m.s. misfit as much as possible while preventing the resistivity model from over fitting the data.**





**Figure C2. Pseudosections of the MT data of Line A rotated to a co-ordinate system with x = N105°E. Each data quantity is compared to the predicted inversion response of inversion 1 with τ = 3.2 after 200 iterations.**





**Appendix D: Re-evaluation of the airborne reflectivity**

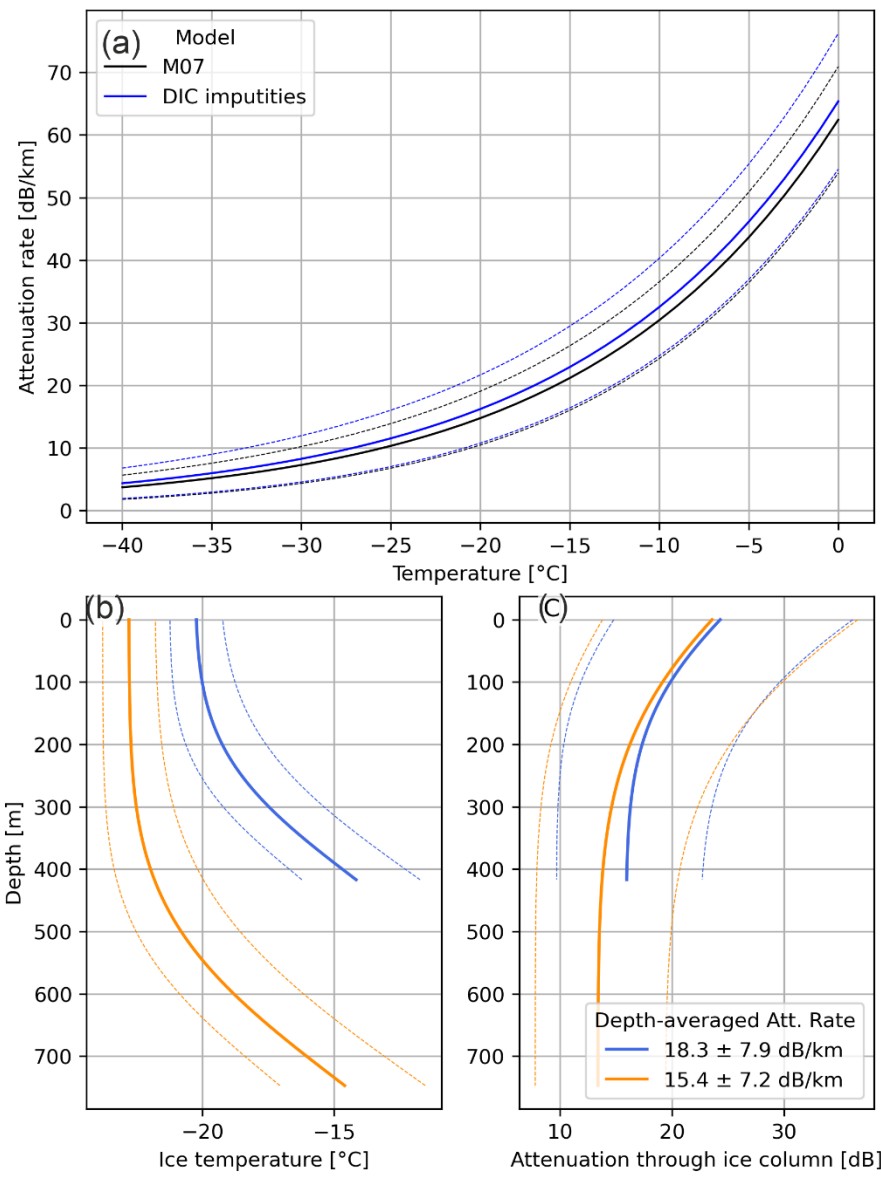

**Figure D1. (a) Attenuation rate as a function of ice temperature from the models used over DIC and Greenland (M07). (b) Example ice temperature profiles including their uncertainty ranges, and (c) resulting attenuation rates (one-way) through the ice column, propagating ice temperature and impurity uncertainties.**



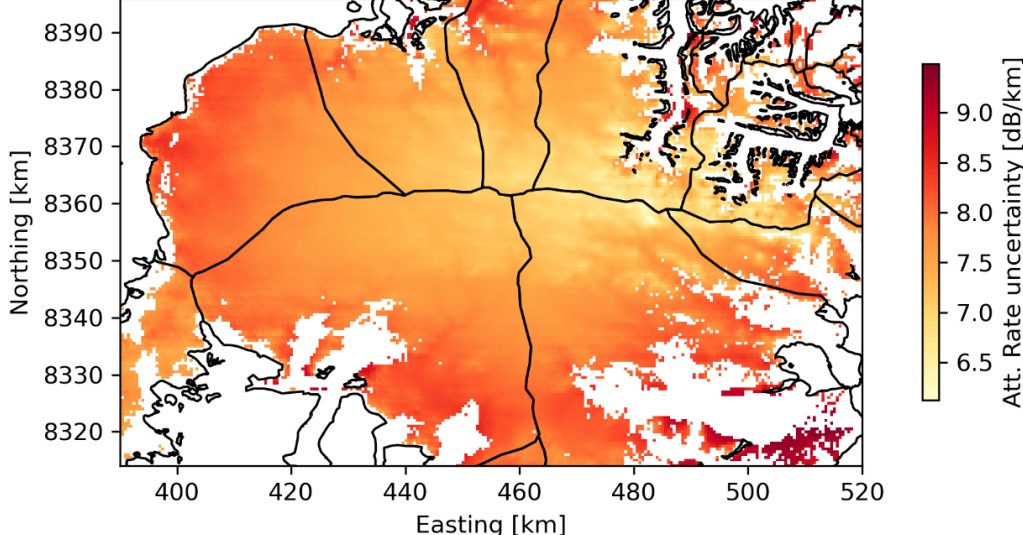


**Figure D2. Uncertainty of Arrhenius modeled attenuation rates over DIC.**



**Table D1. Parameters used in the Arrhenius-type conductivity model to estimate attenuation rates across DIC and over the NW Greenland subglacial lakes.**

| Symbol | Description | Units | Value |
|---|---|---|---|
| $T_r$ | Reference temperature | K | 252[a] |
| $T$ | Ice temperature | K | Modeled using a steady-state 1D advection-diffusion model |
| $\sigma_{pure}$ | Conductivity of pure ice | $\mu Sm^{-1}$ | $9.2 \pm 0.2$[a] |
| $\mu_{H^+}$ | Molar conductivity of $H^+$ | $S\ m^{-1}\ M^{-1}$ | $3.2 \pm 0.5$[a] |
| $\mu_{Cl^-}$ | Molar conductivity of $Cl^-$ | $S\ m^{-1}\ M^{-1}$ | $0.43 \pm 0.07$[a] |
| $\mu_{NH_4^+}$ | Molar conductivity of $NH_4^+$ | $S\ m^{-1}\ M^{-1}$ | 0.8[a] |
| $[H^+]$ | Molar concentration of $H^+$ | $\mu M$ | $1.82 \pm 1.34$[b] / $1.6 \pm 1.2$[c] |
| $[Cl^-]$ | Molar concentration of $Cl^-$ | $\mu M$ | $1.00 \pm 0.82$[b] / $0.4 \pm 0.4$[c] |
| $[NH_4^+]$ | Molar concentration of $NH_4^+$ | $\mu M$ | $1.20 \pm 1.31$[b] / $0.5 \pm 0.6$[c] |
| $E_{pure}$ | Activation energy of pure ice | eV | $0.51 \pm 0.01$[a] |
| $E_{H^+}$ | Activation energy of $H^+$ | eV | $0.20 \pm 0.04$[a] |
| $E_{Cl}$ | Activation energy of $Cl^-$ | eV | $0.19 \pm 0.02$[a] |
| $E_{NH_4^+}$ | Activation energy of $NH_4^+$ | eV | 0.23[a] |

[a] Values taken from the M07 model for the Greenland Ice Sheet as described in (MacGregor et al., 2015) and applied by (Jordan et al., 2016).[b] Average concentration measured along a DIC firn core (Criscitiello et al., 2021), and used for the Arrhenius attenuation rate model over DIC. $H^+$ is derived from the $HNO_3$ concentrations. [c] Impurity concentrations during the Holocene epoch used by (MacGregor et al., 2015), and here used for the Arrhenius attenuation rate model over the NW Greenland sub.

**Data availability**

All seismic, TEM and MT data, acquired on DIC, used in this study are available from https://doi.org/10.5281/zenodo.7641565 (Killingbeck et al., 2023). The SRH1 DIC airborne radar data re-evaluated in this study was accessed from: https://doi.org/10.5281/zenodo.5795105 (Rutishauser et al., 2022). The Operation Ice Bridge radar data over DIC are available on the CReSIS public webpage https://data.cresis.ku.edu/. Impurity concentrations used in Arrhenius temperature attenuation relationship can be accessed at: https://bit.ly/3k6UCua (Criscitiello et al., 2021).

**Author contributions**

Conceptualization: SFK, CFD, MJU, and ADB. Methodology: SFK, CFD, MJU and AR. Software: SFK, MJU and AR. Validation: SFK, JK, MJU and AR. Formal analysis: SFK, AR, MJU, JK, TH, and ADB. Investigation: JK, TH, BM and EB. Resources: SFK, ASC, MJU and AR. Data Curation: SFK. Writing - Original Draft: SFK, AR, AD, MJU, and ASC. Writing - Review & Editing: All authors. Visualization: SFK and AR. Supervision: CFD, ASC and AD. Project administration: SFK, AD and ASC. Funding acquisition: AD, AR, ASC, CFD and MJU.

**Competing interests**

One of the (co-)authors is a member of the editorial board of The Cryosphere, and the authors have no other conflicts of interests to declare.

**Acknowledgments**

We thank the Polar Continental Shelf Program for logistical support throughout the field season; Rob Harris at Geonics for his 510 support and help with the TEM method; Zoe Vestrum at the University of Alberta for her MT support during deployment to the field; Nikolaj Foged at Aarhus University for his support with the TEM data processing; Natalie Wolfenbarger at the University of Texas Institute for Geophysics for useful discussions about RES attenuation rates.

**Financial support**

This research was funded by the Weston Family Foundation. The aircraft hours were funded by the Polar Continental Shelf 515 Program (PCSP) and ArcticNet. MT survey was supported by a NSERC Discovery Grant to Martyn Unsworth and the Future Energy Systems program at the University of Alberta.

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
