# Peer review of "Misidentified subglacial lake beneath the Devon Ice Cap, Canadian Arctic: A new interpretation from seismic and electromagnetic data"

_EGUsphere, 2024_

## Referee Comment (RC2)

**Specific comments:**

**Structure:**

From the introduction it is was not entirely clear to me, that the authors will be re-evaluating the radio echo sounding (RES) data and, therefore, section 4 comes a bit out of nowhere. However, I think that it is relevant and I believe it deserves to be named as an objective of this study directly in the introduction. Additionally, parts of the derivation of the Arrhenius modeled attenuation rates sub-section might be better suited to the methods section. If you follow this comment, I think it may be worth to briefly introduce the RES method, in particular for readers who are not familiar with the method and the previous studies.

**Transient Electromagnetic Method (TEM):**

In Line 143 you refer to Grombacher et al. (2021) for the induced polarization effect. Although, this study is relevant for airborne TEM data in permafrost environments, it is not at the core of induced polarization literature. Maybe you could elaborate a bit on the physical background of the induced polarization effect and a few earlier studies on the presence of the IP effect in TEM data. Please find here a few suggestions that you might find useful (in no particular order):

*Weidelt, 1982; "Response characteristics of coincident loop transient electromagnetic systems"; 10.1190/1.1441393*

*Kozhevnikov and Antonov, 2012; "Fast-decaying inductively induced polarization in frozen ground: a synthesis of results and models"; 10.1016/j.jappgeo.2012.03.008*

*Kang and Oldenburg,, 2016; "On recovering distributed IP information from inductive source time domain electromagnetic data"; 10.1093/gji/ggw256*

In the same sentence (Line 143) you state that you do not invert for IP effects, however you do not really justify this choice. Why did you refrain from inverting for the IP effects in TEM data? Could you maybe add the full raw data curve of your 7 soundings and mark possible negative voltage readings?

**Magnetotelluric Method (MT):**

In the TEM inversion you selected a range of 1e3 Ωm to 1e6 Ωm for the ice layer, yet in the MT you fixed the value for the ice layer at 1e5 Ωm. Why didn't you select the same constraints for the ice layer for MT and TEM?

**Seismic reflection method:**

In Line 192 you state that due to the positive polarity of the ice base interface and the second reflector, it is unlikely that the  material beneath the ice is a lake. Is this only derived from the numerical example in Figure 3b, or do you have a physical explanation for this reasoning?

**Materials below the DIC:**

In sub-section 3.3. you describe the physical properties of the materials below the DIC, but as far as I understood, the  acoustic impedance points to a high clay content of up to 40%, whereas the EM methods solve for an electrical resistivity between 1'000 Ωm and 10'000 Ωm, which contradicts the high clay content of 40% a bit. Could you please clarify this?

**Conclusion:**

In lines 448 – 450 you state that future studies should include a sensitivity analysis of the attenuation-rate. Maybe you could elaborate on this and provide some even more explicit guidelines for future studies to avoid the misinterpretation of a subglacial lake.

**Technical corrections and suggestions:**

Line 74: Active source seismic, TEM and MT data were all collected in the same field campaign

Line 154: I believe the serial numbers of the MT device are not relevant here. Please remove for the sake of conciseness, or justify why they are relevant.

What do you mean exactly by r.m.s (e.g., Line 283)? An absolute , relative or error weighted value for the misfit between modelled and measured data?

For ranges of values and units, I personally prefer to have units at both the lower and upper end of the range (e.g., Line 282). You are actually using the percentages in Line 281 for both values! Please use units for all values consistently throughout the paper.

Line 372: Equation 4 is cut-off at the end of the page.

Throughout the text of the manuscript you are using Ωm, but in a two Figures (5, 10) you are using Ohm.m, while you are using **Ω.m** is the captions. Could you please use Ωm consistently throughout the paper?

---

## Author Response (AR1)

**REVIEWER 1**

Thank you for your time reviewing our manuscript and the suggestion of publication. For the reasons discussed below we believe this manuscript should be published as a standalone review article in TC. Please find our response (AR) to RC1 comments below:

RC1: This paper further explores a region of the Canadian Arctic where a previous radio echo sounding mission had "discovered" a putative hypersaline lake beneath a thick layer of glacial ice. The authors have followed up on the purported discovery with further geophysical investigations including seismic, magnetotellurics and time-domain electromagnetics. All of the new data indicate that the radio sounding detection of a hypersaline lake may have been a misidentification. The new data, including seismic reflectivity and MT/TEM inversions, do not support the interpretation of a hypersaline layer. A refined analysis of radar attenuation rates involving a couple of methods now point toward the absence of a hyper saline layer; such that the authors now conclude that the provocative feature is not likely to exist beneath the glacial ice. The conclusion of the present study highlights the importance of using multiple geophysical methods to follow up on provocative findings based on analysis of a single method. Generally, the work in this paper has been carefully performed and checked.

AR: This research represents a new interpretation of the subglacial environment under Devon Ice Cap, with evidence based on new surface based geophysical techniques that have been applied to this region. The previous published manuscript did not present incorrect data. Rather the authors of that paper presented a different interpretation based on the airborne radar data observations they had available at the time, using a standard method to process the RES data. Therefore, this manuscript is most definitely not a correction to the RES manuscript of Rutishauser at al., (2021). Furthermore, the analysis of the previous airborne radar data observations represents a relatively small portion of this manuscript with most of the focus on our combined analysis of the new active seismic, MT and TEM data that we collected at this site in 2022.

As highlighted by Reviewer 2, the main findings are 1) that the detection of subglacial lakes from RES data is highly dependent on the choice of attenuation rates applied, and attenuation rate sensitivity analysis should be used to determine the uncertainty of RES datasets, 2) that the collection of complimentary surface-based geophysical data sets is essential to reduce ambiguity that is inherent in RES data, and 3) that there is no subglacial lake beneath Devon Ice Cap.

RC1: I do not see any major flaws in the analysis. However, the main finding is simply that a previous study was in error and has now been corrected. As such, despite the obvious large amount of effort the authors have undertaken, the paper does not stand alone as a significant new piece of research. The geophysical methodology is standard and there is not a new discovery being reported herein. As such, I strongly recommend the paper be

published as a "comment" on the original manuscript. This does not diminish its importance to the scientific community compared to how a stand-alone article would be received; in fact, I think publishing it as a correction enhances its visibility and moreover it is the appropriate course of action.

AR: We apologise that the novelty of the geophysical approach and significance of the research was not made clear in the introduction. We will address this in a revised manuscript. The standard geophysical methods used to detect subglacial lakes are RES and seismic reflection. The use of EM methods to characterize subglacial hydrology and lakes is novel within the glaciology community. Although they have been used in other settings/industries. to date there have been no joint seismic, MT and TEM studies for subglacial lake detection. Even though the data do not show evidence of a subglacial lake beneath Devon Ice Cap, the synthetic analysis introduced in this manuscript (Figure 4, Figure 5 and Figure 7) will be highly useful for the glaciology community for future studies investigating the potential presence of subglacial lakes (as commented on by reviewer 2). Further, the radar attenuation rate sensitivity study conducted in this manuscript highlights a best practice, which we encourage the glaciological community to use for RES detection of subglacial lakes in the future.

Science as a discipline is not solely focused on publishing "new discoveries". It also includes developing new understandings and evidence based on the availability of additional data or novel data analysis. An example is the current Mars South Pole debate, where there are a number of published papers in high impact journals support and disagree with the hypothesis of brine deposits on Mars in light of a growing evidence base.

**REVIEWER 2**

Thank you for your time reviewing our manuscript and the valuable detailed comments. Please find our response (AR) to the comments by reviewer 2 (RC2) below:

RC2: From the introduction it is was not entirely clear to me, that the authors will be re-evaluating the radio echo sounding (RES) data and, therefore, section 4 comes a bit out of nowhere. However, I think that it is relevant and I believe it deserves to be named as an objective of this study directly in the introduction. Additionally, parts of the derivation of the Arrhenius modelled attenuation rates sub-section might be better suited to the methods section. If you follow this comment, I think it may be worth to briefly introduce the RES method, in particular for readers who are not familiar with the method and the previous studies.

AR: Thank you for this comment. We have edited the last paragraph of the introduction to include the re-evaluation of the RES data as an objective of the study (lines 72 – 87). To address the second part of this comment we have changed the title of Section 2 to "Ground-based geophysical methods" and added an extra subsection title at the start of

section 4 to introduce the methods for calculating basal reflectivity: "4.1 Methods for calculating basal reflectivity".

RC2: In Line 143 you refer to Grombacher et al. (2021) for the induced polarization effect. Although, this study is relevant for airborne TEM data in permafrost environments, it is not at the core of induced polarization literature. Maybe you could elaborate a bit on the physical background of the induced polarization effect and a few earlier studies on the presence of the IP effect in TEM data. Please find here a few suggestions that you might find useful (in no particular order):

Weidelt, 1982; "Response characteristics of coincident loop transient electromagnetic systems"; 10.1190/1.1441393

Kozhevnikov and Antonov, 2012; "Fast-decaying inductively induced polarization in frozen ground: a synthesis of results and models"; 10.1016/j.jappgeo.2012.03.008

Kang and Oldenburg,, 2016; "On recovering distributed IP information from inductive source time domain electromagnetic data"; 10.1093/gji/ggw256

In the same sentence (Line 143) you state that you do not invert for IP effects, however you do not really justify this choice. Why did you refrain from inverting for the IP effects in TEM data? Could you maybe add the full raw data curve of your 7 soundings and mark possible negative voltage readings?

AR: Thank you for the useful references and comments. We will add Figure B1 to Appendix B showing the measured TEM soundings. We believe the negative received voltages recorded at the early time gates (< 0.1ms) are due to an over saturation, likely from remanent current still in the transmitter loop after the turn off time when the receiver measurement period begins. We removed these negative data and any adjacent positive data which look to be distorted (flatten) from the over saturation (0.1 ms – 0.2 ms). At the late times (> 3 ms), we observe a flip from positive to negative data (similar to that shown in Figure 2 of Weidelt, 1982) highlighting that there may be an IP effect in our data and the likely existence of chargeable material in the subsurface (Weidelt 1982). The observation of an IP effect on our TEM data suggests a high-resistivity subsurface, where the very resistive background heightens the IP response from the Earth (e.g., Grombacher et al. 2021). This observation is clearly contrary to the existence of a conductive hypersaline lake.

In this study we have inverted the positive TEM data for a normal decay curve to give an estimate of the resistivity range of the ice and subglacial material. However, the IP effect on our data has likely distorted the normal decay curve and therefore should be correctly modelled in any future inversions of this dataset to determine the depth and magnitude of the chargeable material in the subglacial environment. We hope to further investigate this in future work. Importantly, the IP effect on our MT data is negligible and since our

TEM and MT measured data are consistent (Figure 5), we feel the results from our TEM inversion are sufficient in providing evidence for the lack of a hypersaline lake beneath DIC.

We will add this text/information to the last two paragraphs in section 2.2.1, lines 149 – 165.

RC2: In the TEM inversion you selected a range of 1e3 Ωm to 1e6 Ωm for the ice layer, yet in the MT you fixed the value for the ice layer at 1e5 Ωm. Why didn't you select the same constraints for the ice layer for MT and TEM?

AR: This was due to the different inversion methods applied to the TEM (stochastic) and MT (deterministic) datasets. Importantly, even though different inversion methods were applied the results from both EM methods are very consistent. Firstly, the constrained Bayesian inversion of the TEM data was conducted with an ice resistivity range of 1e3 Ωm to 1e6 Ωm. Figure 6 shows the results which show the ice resistivity is in the range 1e4 to 1e6 Ωm. Therefore, 1e5 Ωm is the average ice resistivity from the TEM inversion results which we used in the MT inversion to fix the ice resistivity. We have added this to the manuscript. Line 183-184: "The ice was assigned a resistivity of 100000 Ωm (estimated from the results of the TEM Bayesian inversion; Fig. 6) and fixed in the inversion."

RC2: In Line 192 you state that due to the positive polarity of the ice base interface and the second reflector, it is unlikely that the material beneath the ice is a lake. Is this only derived from the numerical example in Figure 3b, or do you have a physical explanation for this reasoning?

AR: This is also derived from the numerical example in Figure 4, which shows that an ice-water interface will always have a negative polarity, an important observation for subglacial lake detection. The physical explanation is that the seismic velocity of water can vary between ~1450 m/s to 1570 m/s, the density varies between 1 and 1.05 g/cm$^3$, both variables are salinity dependant. Whereas the seismic velocity of ice is 3600 – 3800 m/s with density 0.92 g/cm$^3$. The acoustic impedance (AI) is the product of velocity and density. The polarity of the seismic reflection from an interface is dependent on the AI change between the two materials. Ice is acoustically hard and water is acoustically soft (AI ice > AI of water) so there will be a negative reflection. We have edited line 200 for better clarity: "The polarity of R1 is opposite to that expected for subglacial water (Fig. 3b and Fig. 4), indicating the material directly under the ice is unlikely to be a lake."

RC2: In sub-section 3.3. you describe the physical properties of the materials below the DIC, but as far as I understood, the acoustic impedance points to a high clay content of up to 40%, whereas the EM methods solve for an electrical resistivity between 1'000 Ωm and 10'000 Ωm, which contradicts the high clay content of 40% a bit. Could you please clarify this?

AR: Apologies for the confusion here. The lowest error bar of the AI lies just above the upper marker of the less than 40% clay content (< 40% clay), and far away from the greater than 40% clay marker (> 40% clay). We interpret this as a low clay content. We have edited line 331 for clarity: "... suggest the material could have a low clay content and a low content of unfrozen saline pore fluid."

RC2: In lines 448 – 450 you state that future studies should include a sensitivity analysis of the attenuation-rate. Maybe you could elaborate on this and provide some even more explicit guidelines for future studies to avoid the misinterpretation of a subglacial lake.

AR: Thank you for this comment, we will add text to the final paragraph of the conclusion to address this comment (Line 449). "We suggest the current criteria for subglacial lake detection from RES datasets includes an attenuation rate sensitivity assessment to quantifying uncertainty in the basal reflectivity. Furthermore, we suggest a move towards reporting RES proposed subglacial lakes probabilistically or using confidence levels, e.g., Bowling et al., 2019. Finally, our study highlights that the acquisition of multiple geophysical techniques, where logistically possible, are essential to reliably interpret subglacial water systems."

At this stage it is beyond the scope of this paper to provide more explicit guidelines for detecting subglacial lakes using RES alone. However, this study has highlighted the current criteria needs to be re-assessed and this is part of future work our team is looking into addressing.

Technical corrections and suggestions:

RC: Line 74: Active source seismic, TEM and MT data were all collected in the same field campaign

AR: Changed to "Here, active source seismic, TEM and MT data were collected in the same field campaign."

RC: Line 154: I believe the serial numbers of the MT device are not relevant here. Please remove for the sake of conciseness, or justify why they are relevant.

AR: These have been removed.

RC: What do you mean exactly by r.m.s (e.g., Line 283)? An absolute, relative or error weighted value for the misfit between modelled and measured data?

AR: Error weighted value This is r.m.s. misfit and is a normalised statistical measure of the data misfit. An ideal fit would result in a value of 1

RC: For ranges of values and units, I personally prefer to have units at both the lower and upper end of the range (e.g., Line 282). You are actually using the percentages in Line 281 for both values! Please use units for all values consistently throughout the paper.

AR: This has been addressed throughout the manuscript.

RC: Line 372: Equation 4 is cut-off at the end of the page.

AR: this has been fixed

RC: Throughout the text of the manuscript you are using Ωm, but in a two Figures (5, 10) you are using Ohm.m, while you are using Ω.m is the captions. Could you please use Ωm consistently throughout the paper?

AR: This has been fixed.

**REVIEWER 3**

Thank you for your time reviewing our manuscript and the minor comments. Please find our response (AR) to reviewer 3 (RC3) comments below:

RC3: There is no mention of bathymetry or topography data. Can you indicate if the seismic datasets provided subglacial terrain profiles used for the models, such as in the MT inversions? (Fig. 9)

AR: We used the subglacial topography data from the detailed RES survey presented in Rutishauser et al. 2021. Bathymetry was not included owing to the distance to sea.

RC3: A sentence or two on the significance of the presence of a subglacial lake would be worthwhile for non-glaciology readers. It is stated that a DIC subglacial lake would have been the first found in the Canadian Arctic. Why would this have been so profound to justify this field campaign that was (I assume) to elaborate on, rather than validate, the results from the 2011 - 2015 RES study? This is a highly remote region requiring complex logistics to access, therefore there must be a strong reason to conduct further study on the DIC subglacial lake.

AR: The next phase of this research project was to drill and sample the subglacial water. Hence, a detailed ground based geophysical survey was required to decide the best site for drilling (see Line 73 of the introduction). However, since our results do not provide any evidence of a subglacial lake this drilling phase has been cancelled. We will add this information into a revised manuscript.

RC3: Is it possible that there is a temporal component to the presence of a hypersaline subglacial lake? Such that, increased glacial melt in the summer leads to the formation of a subglacial lake, which then refreezes in the colder months. Stating when in the year this geophysical campaign took place may close this line of inquiry.

AC: At this elevation (summit of DIC ~1800 m.a.s.l.), there is minimal surface melt and no crevasses, so it is unlikely meltwater is routed to the glacier bed. Furthermore, the repeat laser altimetry does not show any significant surface elevation changes (besides what we would expect from mass balance). Additionally, the bed is expected to be

below the pressure melting point all year, so we would not expect any change in salinity/melting/refreezing. Finally, the ice flow is extremely slow in this area and the ice is thought to be frozen to the bed. All geophysical surveys (including the detailed RES survey) were acquired in spring. Comment will be added to the manuscript.

RC3: In summary, this paper is worthwhile for publication, primarily as a demonstration of coeval geophysical studies of subglacial environments, secondarily as an argument for multiple studies to mitigate against false results.

AC: Thank you for the recommendation of publication.

**LIST OF CHANGES IN THE MANUSCRIPT**

Added sentence at line 75 to highlight the innovative aspects of our manuscript.

Edited the last paragraph of the introduction to include the re-evaluation of the RES data as an objective of the study (lines 72 – 87).

Changed the title of Section 2 to "Ground-based geophysical methods" and added an extra subsection title at the start of section 4 to introduce the methods for calculating basal reflectivity: "4.1 Methods for calculating basal reflectivity".

Added Figure B1 to Appendix B showing the measured TEM soundings.

Edited text in TEM section (Line 151) to discuss IP effect on soundings.

Added text (Line 473) to conclusions to elaborate on this and provide some even more explicit guidelines for future studies to avoid the misinterpretation of a subglacial lake.

Addressed all technical corrections and suggestions.

---

## Author Response (AR2)

Dear Adrian Flores Orozco,

Thank you for your time reviewing our manuscript and the valuable detailed comments in the attached PDF. Please find our response below:

*Editors' comments: I was able to find many redundancies in your paper.*

Authors response: We have removed all the texted suggested by the editor in the attached pdf.

*Editors' comments: I also believe that the structure of the paper could be improved. In particular, I think you should try to explain as much as possible your models and processes in the inversion and modeling of the data in the Material and Methods (M&M) sections. This would help to reduce the length of the Results section and present there a clear presentation of the outcome of your analysis. Also re-structuring your paper could help to the readers to clearly understand the details of your numerical modeling , and data handling. For instance, the entire sub-section on the estimation of attenuation rates using the Arrhenius-model could be move to M&M and provide there all details.*

Authors response: We have re-structed the manuscript as suggested by the editor and moved the radar attenuation rate analysis section to the methods section.

*Editors' comments: In the attached PDF, I provide a few examples of sections that can be summarized, or re-structured, also a few lines that seems to be redundant. I encourage you to revise your manuscript and maybe re-structure it. Maybe it also helps to reduce its length.*

Authors response: We have revised the manuscript including all the edits and corrections suggested by the editor. We have also moved Figure 12 to the appendix to reduce the manuscripts length.

*Editor comment [in PDF]: it would be convenient to explain in material and methods, how you define the fluid conductivity and ice content for different types of water/temperature*

Authors response: Here, the acoustic properties of the brine at different temperatures have come directly from the results of an acoustic pulse transmission experiment conducted in Prasad and Dvorkin (2004). This is stated in the caption to Figure 4.

*Editors' comment [in PDF]: I believe that the issue of IP effects in EM data needs to be addressed within the discussion and to be mentioned as a further topic of research. It would be interesting to make the TEM data available for the community dealing with IP effects in frozen ground, as expected in TC.*

Authors response: The TEM data is openly available to the community at https://doi.org/10.5281/zenodo.7641565 and we welcome any further studies using this dataset. This link is included in the Data availability section. During TEM processing we

did purchase a SPIA license and run some initial IP inversions, we found the SPIA resistivity inverted result to be very similar to the Bayesian inversion result. We report the Bayesian result in this manuscript as the code is open source and freely available online for the community to use. We do not believe the results from an IP inversion would change our interpretation. Furthermore, we are confident in our EM inversion results as they are consistent with the results from the MT inversions. A detailed IP inversion would enable an estimate of the chargeable material in the subglacial environment; however, we believe this is out of the scope of this manuscript. For these reasons and to avoid confusion to the reader, this paragraph has been removed from the manuscript.